# Discovering Generalizable Multi-agent Coordination Skills from Multi-task Offline Data

**Fuxiang Zhang**[1,2]*, **Chengxing Jia**[1,2]*, **Yi-Chen Li**[1], **Lei Yuan**[1,2], **Yang Yu**[1,2], **Zongzhang Zhang**[1]†

[1]National Key Laboratory for Novel Software Technology, Nanjing University
[2]Polixir Technologies
{zhangfx,jiacx,liyc,yuanl}@lamda.nju.edu.cn
{yuy,zzzhang}@nju.edu.cn

## Abstract

Cooperative multi-agent reinforcement learning (MARL) faces the challenge of adapting to multiple tasks with varying agents and targets. Previous multi-task MARL approaches require costly interactions to simultaneously learn or fine-tune policies in different tasks. However, the situation that an agent should generalize to multiple tasks with only offline data from limited tasks is more in line with the needs of real-world applications. Since offline multi-task data contains a variety of behaviors, an effective data-driven approach is to extract informative latent variables that can represent universal skills for realizing coordination across tasks. In this paper, we propose a novel Offline MARL algorithm to Discover coordInation Skills (ODIS) from multi-task data. ODIS first extracts task-invariant coordination skills from offline multi-task data and learns to delineate different agent behaviors with the discovered coordination skills. Then we train a coordination policy to choose optimal coordination skills with the centralized training and decentralized execution paradigm. We further demonstrate that the discovered coordination skills can assign effective coordinative behaviors, thus significantly enhancing generalization to unseen tasks. Empirical results in cooperative MARL benchmarks, including the StarCraft multi-agent challenge, show that ODIS obtains superior performance in a wide range of tasks only with offline data from limited sources.

## 1 Introduction

Cooperative multi-agent reinforcement learning (MARL) has drawn broad attention in addressing problems like video games, sensor networks, and autopilot (Peng et al., 2017; Cao et al., 2013; Gronauer & Diepold, 2022; Yun et al., 2022; Xue et al., 2022b). Since recent MARL methods mainly focus on learning policies in one single task with simulating environments (Sunehag et al., 2018; Rashid et al., 2020; Lowe et al., 2017; Foerster et al., 2018), there exist two obstacles when applying them to real-world problems. One is poor generalization when facing tasks with varying agents and targets, where the practical demand is to adapt to multiple tasks rather than learning every new task from scratch (Omidshafiei et al., 2017). The other is potentially high costs and risks caused by real-world interactions through an under-learning policy (Levine et al., 2020).

Multi-agent systems are expected to perform flexibly among multiple general scenarios where the agents and targets may differ. Multi-task MARL is one promising way to realize such flexibility and generalizability. Previous related works mainly focus on training simultaneously in a pre-defined task set (Omidshafiei et al., 2017; Iqbal et al., 2021) or fine-tuning a pre-trained policy to target tasks (Hu et al., 2021; Zhou et al., 2021; Qin et al., 2022a) in an online manner. Although these approaches exhibit promising performance in some tasks, the expensive cost of online interactions hinders their applications to a broader range of tasks. Offline RL (Levine et al., 2020), aiming at learning policies from a static dataset, is anticipated to remove the need for interactions during training. However, most current offline RL methods conservatively regularize the learned policies towards datasets (Wu et al., 2019; Kumar et al., 2019; Yang et al., 2021; Fujimoto et al., 2019). Albeit conservatism

---

*These authors contributed equally. Work was done during an internship at Polixir Technologies.
†Zongzhang Zhang is the corresponding author.

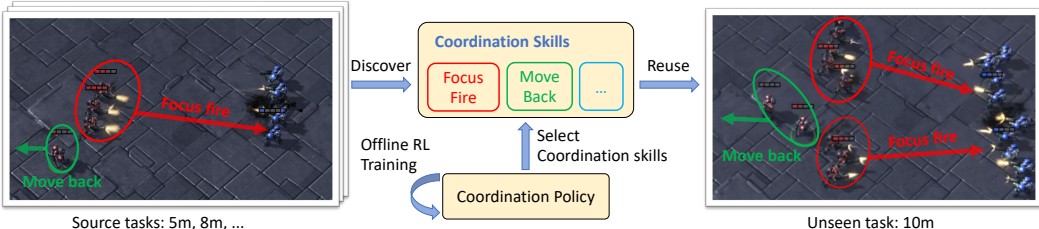

Figure 1: An illustration of coordination skill discovery from multi-task offline data. Offline data from marine battle source tasks like 5m and 8m of StarCraft multi-agent challenge contain generalizable coordination skills like focusing fire, moving back, etc. After discovering these coordination skills from source data, a coordination policy learns to appropriately choose coordination skills through an offline RL training process. When facing the unseen task 10m, the agents reuse the discovered coordination skills to achieve coordination and accomplish the task.

effectively mitigates the distribution shift issue of offline learning, it will restrict the learned policy to be similar to the behavior policy, leading to severe degradation of generalization when facing unseen data (Chen et al., 2021b). Therefore, leveraging multi-agent offline data to train a policy with adequate generalization ability across tasks is in demand.

This paper finds that the underlying generic skills can greatly help to improve the policy's generalization. Indeed, humans are good at summarizing skills from several tasks and reusing these skills in other similar tasks. Taking Figure 1 as an example, we try to learn a policy from some StarCraft multi-agent challenge (Samvelyan et al., 2019) tasks, 5m and 8m, where we need to control five or eight marines separately to beat the same number of enemy marines. Moreover, we aim to directly deploy the learned policy without fine-tuning to an unseen target task, 10m, a task with ten marines on each side. One effective way to achieve this is to extract skills from the source tasks, like focusing fire on the same enemy or moving back low-health units, and then apply these skills to the target task. Although these tasks have different state spaces or action spaces, these skills can be applicable to a broad range of tasks. We refer to such task-invariant skills as *coordination skills* since they are beneficial to realize coordination in different tasks. In other words, extracting such coordination skills from known tasks facilitate generalization via reusing them in unseen tasks.

Towards learning and reusing coordination skills in a data-driven way, we propose a novel Offline MARL algorithm to Discover coordInation Skills (ODIS), where agents only access a multi-task dataset to discover coordination skills and learn generalizable policies. ODIS first extracts task-invariant coordination skills that delineate agent behaviors from a coordinative perspective. These shared coordination skills will help agents perform high-level decision-making without considering specific action spaces. ODIS then learns a *coordination policy* to select appropriate coordination skills to maximize the global return via the centralized training and decentralized execution (CTDE) paradigm (Oliehoek et al., 2008). Finally, we deploy the coordination policy directly to unseen tasks. Our proposed coordination skill is noteworthy compared to previous works in online multi-agent skill discovery (Yang et al., 2020; He et al., 2020), which utilize hierarchically learned skills to improve exploration and data efficiency. Empirical results show that ODIS can learn to choose proper coordination skills and generalize to a wide range of tasks only with data from limited sources. To the best of our knowledge, it is the first attempt towards unseen task generalization in offline MARL.

## 2 RELATED WORK

**Multi-task MARL**. Multi-task RL and transfer learning in MARL can improve sample efficiency with knowledge reuse (da Silva & Costa, 2021). The knowledge reuse across multiple tasks may be impeded by varying populations and input dimensions, asking for policy networks with flexible structures like graph neural networks (Agarwal et al., 2020) or self-attention mechanisms (Hu et al., 2021; Zhou et al., 2021). Recent works consider utilizing policy representations or agent representations to realize multi-task adaptations (Grover et al., 2018). EPL (Long et al., 2020) introduces an evolutionary-based curriculum learning approach to scale up the number of agents. REFIL (Iqbal et al., 2021) adopts randomized entity-wise factorization for multi-task learning. UPDeT

(Hu et al., 2021) utilizes transformer-based value networks to realize adaptations among alterable populations and inputs. However, these approaches only consider simultaneous learning or fine-tuning in different tasks. Training generalizable policies for deployment in unseen tasks remains a challenge.

**Offline MARL**. Offline RL attracts tremendous attention for its data-driven training paradigm without interactions with the environment (Levine et al., 2020). Offline MARL is a promising research direction (Zhang et al., 2021; Formanek et al., 2023) that trains policies from a static dataset. Previous work (Fujimoto et al., 2019) discusses the distribution shift issue in offline learning and considers learning behavior-constrained policies to relieve extrapolation error from unseen data estimations (Wu et al., 2019; Kumar et al., 2019). Existing offline MARL methods often try to adopt conservative constraints upon current online MARL methods, which either extend policy gradient algorithms to multi-agent cases (Lowe et al., 2017; Foerster et al., 2018; Iqbal & Sha, 2019; Wang et al., 2021b; Xue et al., 2022a) or adopt Q-learning paradigms with value decomposition (Sunehag et al., 2018; Rashid et al., 2020; Wang et al., 2021a; Cao et al., 2021; Yuan et al., 2022b;a). Most offline MARL methods consider training policies with sufficient conservatism (Yang et al., 2021; Jiang & Lu, 2021; Pan et al., 2022; Guan et al., 2023). These methods can help learn an effective policy from offline data, but the conservative learning manner may significantly degrade the performance when facing unseen tasks, as learned policies will fail to generalize with out-of-distribution inputs. Some data sharing methods in single-agent RL (Li et al., 2020; Yu et al., 2021; 2022) consider properly using multi-task data for offline policy training. However, potential coordination skills from multi-task MARL data cannot be efficiently exploited by these single-agent data sharing methods.

**Skill discovery with hierarchical structures.** Hierarchical RL (Barto & Mahadevan, 2003; Tang et al., 2019; Pateria et al., 2021) provides an approach to realizing temporal abstraction and hierarchical organization. Skill discovery methods adopt the hierarchical structure to discover unsupervised high-level skills with state empowerment from information theory (Eysenbach et al., 2019; Campos et al., 2020; Sharma et al., 2020). This diverse skill discovery methods from online RL can also be extended to MARL. HSD (Yang et al., 2020) learns latent skill variables via centralized training and MASD (He et al., 2020) considers performing skill discovery in an adversarial way. However, these methods are usually tailored to address data efficiency in online RL, where the proposed skills are different from our multi-task reusable skills towards generalizable MARL. On the other hand, recent works in offline RL also exhibit that high-level latent actions help tackle extrapolation errors from out-of-distribution actions and learn primitive behaviors (Zhou et al., 2020; Ajay et al., 2021). A data-driven approach to extracting and reusing high-level multi-agent behaviors (i.e., our proposed coordination skills) can be a practical and promising direction toward multi-task generalization.

## 3 BACKGROUND

### 3.1 COOPERATIVE MULTI-AGENT REINFORCEMENT LEARNING

A cooperative multi-agent task can be modeled as a decentralized partially observable Markov decision process (Dec-POMDP) (Oliehoek & Amato, 2016) $\mathcal{T} = \langle \mathcal{I}, \mathcal{S}, \mathcal{A}, P, \Omega, O, R, \gamma \rangle$, where $\mathcal{I} = \{1, \ldots, n\}$ is the set of agents, $\mathcal{S}$ is the set of global states, $\mathcal{A}$ is the set of actions, and $\Omega$ is the set of observations. At each time step with state $s \in \mathcal{S}$, each agent $i \in \mathcal{I}$ only acquires an observation $o_i \in \Omega$ drawn from the observation function $O(s, i)$, and then chooses its action $a_i \in \mathcal{A}$. The joint action $\boldsymbol{a} = (a_1, \ldots, a_n)$ leads to a next state $s' \sim P(s' \mid s, \boldsymbol{a})$ and the corresponding global reward $r = R(s, \boldsymbol{a})$. The target is to find a joint policy $\boldsymbol{\pi}(\boldsymbol{a} \mid \boldsymbol{\tau})$ to maximize the discounted return $Q^{\boldsymbol{\pi}}(\boldsymbol{\tau}, \boldsymbol{a}) = \mathbb{E}\left[\sum_{t=0}^{\infty} \gamma^t R(s_t, \boldsymbol{a}_t) \mid s_0 = s, \boldsymbol{a_0} = \boldsymbol{a}, \boldsymbol{\pi}\right]$, where $\gamma \in [0, 1)$ is a discount factor that trades off between current and future rewards. Here $\boldsymbol{\tau} = (\tau_1, \ldots, \tau_n)$, where $\tau_i$ denotes the trajectory $(o_i^1, a_i^1, \ldots, o_i^{t-1}, a_i^{t-1}, o_i^t)$ of agent $i$.

Most value-based cooperative MARL algorithms apply the CTDE paradigm (Sunehag et al., 2018; Rashid et al., 2020; Wang et al., 2021a), where agents can learn a decomposable global value function $Q_{\text{tot}}(\boldsymbol{\tau}, \boldsymbol{a})$ represented by a mixing network in the training phase and use the decomposed value function $Q_i(\tau_i, a_i)$ for decentralized decision. The global value function parameterized by $\theta$ can be learned by minimizing the squared temporal difference (TD) error (Sutton & Barto, 2018), using experience replay and a target network parameterized by $\theta^-$ to stabilize training (Mnih et al., 2015):

$$\min_{\theta} \mathbb{E}_{\boldsymbol{\tau}, \boldsymbol{a}, r, \boldsymbol{\tau}'}\left[\left(r + \gamma \max_{\boldsymbol{a}'} Q_{\text{tot}}\left(\boldsymbol{\tau}', \boldsymbol{a}'; \theta^-\right) - Q_{\text{tot}}(\boldsymbol{\tau}, \boldsymbol{a}; \theta)\right)^2\right].$$

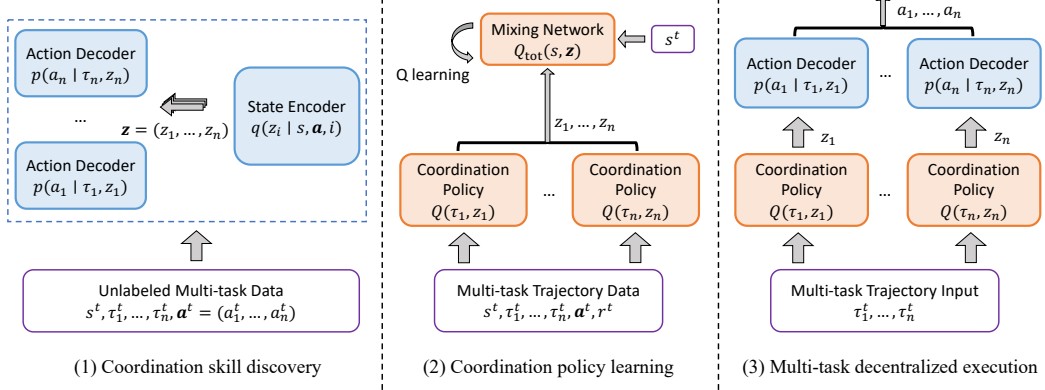

Figure 2: The overall framework of ODIS. ODIS is trained with multi-task offline data from limited tasks and can be deployed to different tasks with varying agents and targets. (1) ODIS learns a state encoder and an action decoder from unlabeled multi-task data (without reward information) to discover coordination skills across different tasks. (2) ODIS performs centralized training to learn coordination policies that can maximize the global return with the help of a mixing network. (3) In multi-task decentralized execution, ODIS individually chooses a coordination skill according to the coordination policy and decodes specific actions using the action decoder.

## 3.2 MULTI-TASK MULTI-AGENT REINFORCEMENT LEARNING

Recent multi-task MARL works consider policy learning among two or several cooperative multi-agent tasks (Hu et al., 2021; Iqbal et al., 2021). In our settings, we focus on learning and deploying policies in a static task set $\{\mathcal{T}_i\}$. A major difficulty in multi-agent multi-task transfer is varying agent numbers and observation/action dimensions. To learn a transferable or universal policy across different tasks, previous methods (Hu et al., 2021) develop a population-invariant network structure by utilizing the transformer structure (Vaswani et al., 2017). For the individual Q-network, the observation $o_i$ of agent $i$ can be decomposed into self/environmental information $o_i^{\mathrm{own}}$ and other entities' information $\{o_{i,j}^{\mathrm{other}}\}$. The network further generates embeddings of $Q, K, V$ and calculate the attention output according to the attention mechanism:

$$Q = \mathrm{MLP}_Q([o_i^{\mathrm{own}}, o_{i,1}^{\mathrm{other}}, \dots]), \; K = \mathrm{MLP}_K([o_i^{\mathrm{own}}, o_{i,1}^{\mathrm{other}}, \dots]), \; V = \mathrm{MLP}_V([o_i^{\mathrm{own}}, o_{i,1}^{\mathrm{other}}, \dots]),$$

$$[e_i^{\mathrm{own}}, e_{i,1}^{\mathrm{other}}, \dots] = \mathrm{softmax}\left(\frac{QK^T}{\sqrt{d_K}}\right)V, \quad d_K = \dim(K).$$

The output representations $e_i^{\mathrm{own}}, e_{i,1}^{\mathrm{other}}, \dots$ can be further utilized to derive Q-values, where $e_i^{\mathrm{own}}$ can be used for actions without interactions with other entities and $e_{i,j}^{\mathrm{other}}$ can be used for interactive actions with entity $j$. To handle the partial observability, we can append an additional historical embedding $h_i^{t-1}$ from the last time step $t-1$ to the transformer input sequence and acquire corresponding $h_i^t$. It is notable that this decomposition technique with transformer models can also be applied to process the state information in mixing networks or other modules (Zhou et al., 2021; Qin et al., 2022a) to effectively handle multi-task inputs with varying data shapes.

## 4 METHOD

In this section, we will describe our proposed Offline MARL algorithm to Discover coordInation Skills (ODIS) from a static multi-task dataset $\mathcal{D}$, which is a fully data-driven approach to learning generalizable policies for multiple tasks. As shown in Figure 2, ODIS begins with unsupervised coordination skill discovery that leverages a state encoder $q(z_i \mid s, \boldsymbol{a}, i)$ to extract the coordination skill $z_i, \dots, z_n$ for each agent using the unlabeled portion of $\mathcal{D}$ without reward information. The discovered coordination skill $z_i$ represents an abstraction of coordinative behaviors, which can be decoded to a task-related action with local information $\tau_i$ by an individual action decoder $p(a_i \mid \tau_i, z_i)$. Both the state encoder and the action decoder are trained in an unsupervised learning manner to

discover effective and distinct coordination skills. After that, ODIS learns a coordination policy to select appropriate coordination skills with offline RL training in the entire dataset $\mathcal{D}$. This offline MARL training process adopts the CTDE paradigm with a conservative learning manner. In the decentralized execution phase, the agent can use individual coordination policy to choose the best coordination skill and decode it to a specific action with the pre-trained action decoder. The high-level decision-making of ODIS promotes generalization across different tasks, especially in unseen tasks with varying targets and controllable agents. To tackle the alterable data shapes from multi-task settings, we design flexible and powerful network structures based on the transformer model to process input data, whose details are illustrated in Appendix C.

## 4.1 Coordination Skill Discovery

Coordination skill discovery is expected to extract effective coordination skills from offline data. We assume the coordination skill $z_i$ for agent $i$ is a discrete variable from a finite set $\mathcal{Z}$, where $|\mathcal{Z}|$ is a hyper-parameter varying among tasks. We leverage a state encoder $q(z_i \mid s, \boldsymbol{a}, i)$ using the sampled state $s$ and joint action $\boldsymbol{a}$ to obtain a coordination skill $z_i$ for agent $i$. By feeding states and joint actions as input, we can ensure the state encoder outputs appropriate coordination skills from the global perspective. In practice, we implement the state encoder with a transformer model, which simultaneously outputs coordination skills $\boldsymbol{z} = (z_1, \ldots, z_n)$ of all agents in a sequence. Moreover, we need to derive task-related actions from the coordination skill in decentralized executions, where acquiring global information is impractical for individual agents. Thus we further introduce an action decoder that can predict a task-related action $\hat{a}_i \sim p(\cdot \mid \tau_i, z_i)$ with an agent $i$'s local information $\tau_i$ and the chosen coordination skill $z_i$. The training objective is to maximize the likelihood of the real action $a_i$ from data, along with the Kullback-Leibler divergence between $q(z_i \mid s, \boldsymbol{a}, i)$ and an uniform prior $\tilde{p}(z_i)$ as a regularization, following $\beta$-VAE (Higgins et al., 2017). The regularization with a uniform distribution of coordination skills can prevent the state encoder from choosing similar coordination skills for all inputs, thereby helping to discover distinguished coordination skills. The objective for coordination skill learning is presented as follows:

$$L_{\mathrm{s}}(\theta_a, \phi_s) = -\mathbb{E}_{(s,\boldsymbol{\tau},\boldsymbol{a})\sim\mathcal{D}} \left[ \sum_{i=1}^{n} \mathbb{E}_{z_i\sim q(\cdot|s,\boldsymbol{a},i)} \left[\log p(a_i \mid \tau_i, z_i)\right] - \beta D_{\mathrm{KL}} \left( q(\cdot \mid s, \boldsymbol{a}, i) \parallel \tilde{p}(\cdot) \right) \right],$$
(1)

where $\phi_s$ and $\theta_a$ denote the parameters of the state encoder and the action decoder, respectively, and $\beta$ is the regularization coefficient. Our implementations of the state encoder and the action decoder utilize the transformer structure to handle alterable dimensions of states, observations, and actions in multi-task settings. The key technique is to decompose input into sequence data as illustrated in Section 3.2 and project each portion to an embedding with a fixed dimension. In Appendix C we provide detailed descriptions of the network structures.

## 4.2 Coordination Policy Learning

After coordination skill discovery, we train an action decoder that can generate a task-related action using an agent's local information and input coordination skills. Therefore, we can further consider developing high-level decision-making on discovered coordination skills. As our coordination skills are universal for a given task set, we can train a generalizable coordination policy to decide on appropriate coordination skills for different tasks.

We adopt a value-based MARL method with the CTDE paradigm to learn a global value function $Q_{\mathrm{tot}}(\boldsymbol{\tau}, \boldsymbol{z})$ that can be decomposed into individual value functions $Q_1(\tau_1, z_1), \ldots, Q_n(\tau_n, z_n)$. The global value function $Q_{\mathrm{tot}}(\boldsymbol{\tau}, \boldsymbol{z})$ is trained with the squared TD loss as follows:

$$L_{\mathrm{TD}}(\theta_v) = \mathbb{E}_{(s,\boldsymbol{\tau},\boldsymbol{a},r,\boldsymbol{\tau}')\sim\mathcal{D},\boldsymbol{z}} \left[ \left( r + \gamma \max_{\boldsymbol{z}'} Q_{\mathrm{tot}} \left( \boldsymbol{\tau}', \boldsymbol{z}'; \theta_v^- \right) - Q_{\mathrm{tot}}(\boldsymbol{\tau}, \boldsymbol{z}; \theta_v) \right)^2 \right],$$
(2)

where we use $\theta_v$ to denote all parameters in value networks including individual and mixing networks, $\theta_v^-$ to denote parameters of target networks. Note that in Equation 2, we cannot directly acquire the joint coordination skill $\boldsymbol{z}$ from the offline dataset $\mathcal{D}$. Therefore, the state encoder $q(z_i \mid s, \boldsymbol{a}, i)$ trained in the coordination discovery phase is reused to provide a joint coordination skill $\boldsymbol{z} = (z_1, \ldots, z_n)$ from the state and joint action. When estimating Q-targets, we choose the joint coordination skill $\boldsymbol{z}'$ by selecting each coordination skill $z_i'$ with maximal individual Q-value $Q_i(\tau_i', z_i')$ to avoid search in

the large joint coordination skill space, following previous MARL methods (Sunehag et al., 2018; Rashid et al., 2020). When the value decomposition method satisfies the individual-global-max (IGM) principle (Wang et al., 2021a), the action selection with individual value functions is correct. We adopt a QMIX-style mixing network to ensure that it can satisfy this property.

Our design of the individual value network and mixing network also utilizes the powerful transformer structure. Practically, the individual value network calculates local Q-values by taking representations from an observation encoder as input. The observation encoder has the same network structure as the aforementioned action decoder to process local trajectory information. The mixing network takes the global state information as input to generate non-negative weights for combining individual Q-values. We also provide detailed descriptions of these networks in Appendix C.

As the coordination skills from the previous discovery phase are obtained with global information, it remains a challenge that the coordination policy may not explicitly learn to choose effective coordination skills only with local observations. We find that directly learning this from the TD objective will be extremely inefficient. Therefore, it will be necessary to introduce an auxiliary objective to learn better representations that can guide the coordination policy towards a coordinative perspective. When updating parameters of the observation encoder, we use the last layer of the previous state encoder to process output representations from the observation encoder and thus acquire a coordination skill distribution $\hat{q}_i(\cdot \mid \tau_i)$. We expect that the output distribution can be similar to the pre-trained state encoder $q(\cdot \mid s, \boldsymbol{a}, i)$. We calculate the KL-divergence between them to update the observation encoder as the consistent loss $L_{\mathrm{c}}$ below:

$$L_{\mathrm{c}}(\phi_o) = \sum_{i=1}^{n} \mathbb{E}_{(s,\boldsymbol{\tau},\boldsymbol{a})\sim\mathcal{D}} \left[ D_{\mathrm{KL}} \left( \hat{q}_i(\cdot \mid \tau_i) \parallel q(\cdot \mid s, \boldsymbol{a}, i) \right) \right], \quad (3)$$

where $\phi_o$ denotes parameters of the observation encoder in individual value network.

To tackle the out-of-distribution issue in offline RL, we also adopt the popular conservative Q-learning (CQL) method (Kumar et al., 2020). To be concise, the total loss term in the coordination exploitation phase is presented as $L_{\mathrm{p}}(\theta_v, \phi_o) = L_{\mathrm{TD}}(\theta_v) + \alpha L_{\mathrm{CQL}}(\theta_v) + \lambda L_{\mathrm{c}}(\phi_o)$, where $\alpha$ and $\lambda$ are two coefficients and $L_{\mathrm{CQL}}$ is the loss term from CQL.

**Decentralized Execution.** When performing decentralized executions in a test task, we use local information to calculate Q-values for each coordination skill with individual value network $Q_i(\tau_i, z_i)$ and then choose the coordination skill with maximal Q-value. The action decoder further uses the coordination skill and local information to provide a task-related action for the particular task.

## 5 EXPERIMENTS

In this section, we design experiments to evaluate the following properties of ODIS[1]. (a) The ability of multi-task generalization, including zero-shot generalization in unseen tasks. We conduct experiments in specially designed task sets from the StarCraft multi-agent challenge (SMAC) (Samvelyan et al., 2019) using offline data with diverse qualities. (b) The semantics of discovered coordination skills. We analyze the coordination skill usage in several test episodes from different tasks to investigate how ODIS makes decisions with different coordination skills. (c) Effectiveness of the ODIS structure. We conduct ablation studies to find out how the components of ODIS affect performance.

### 5.1 PERFORMANCES ON MULTI-TASK GENERALIZATION

**Baselines.** We introduce comparable baselines and perform several adaptations since none of the existing multi-task MARL algorithms considers offline learning. UPDeT (Hu et al., 2021) is a state-of-the-art multi-agent transfer learning baseline that adopts a transformer-based individual network to tackle multi-agent transfer. However, the mixing network of UPDeT is not designed for simultaneous multi-task learning. As its alternatives, we implement two variants of UPDeT by adopting the transformer-based mixing network of ODIS (**UPDeT-m**), and the linear decomposable network of VDN (Sunehag et al., 2018) (**UPDeT-l**), respectively. We also keep the same transformer structures and adopt the same CQL loss to these UPDeT baselines. Therefore, UPDeT-m can be

---

[1]Code available at `https://github.com/LAMDA-RL/ODIS`

Table 1: Average test win rates of the final policies in the task set marine-hard with different data qualities. The listed performance is averaged over five random seeds. We abbreviate asymmetric task names for simplicity. For example, the task name "5m6m" denotes the SMAC map "5m_vs_6m". Results of BC-best stands for the best test win rates between BC-t and BC-r.

| Task | Expert | | | | Medium | | | |
|------|--------|--------|--------|-------------|--------|--------|--------|-------------|
| | BC-best | UPDeT-l | UPDeT-m | ODIS (ours) | BC-best | UPDeT-l | UPDeT-m | ODIS (ours) |
| | | | | Source tasks | | | | |
| 3m | 97.7 ± 2.6 | 71.0 ± 16.6 | 82.8 ± 16.0 | **98.4 ± 2.7** | 65.4 ± 14.7 | 56.6 ± 14.2 | 51.2 ± 3.4 | **85.9 ± 10.5** |
| 5m6m | 50.4 ± 2.3 | 12.1 ± 12.6 | 17.2 ± 28.0 | **53.9 ± 5.1** | 21.9 ± 3.4 | 5.6 ± 4.8 | 6.3 ± 4.9 | **22.7 ± 7.1** |
| 9m10m | **95.3 ± 1.6** | 26.6 ± 12.0 | 3.1 ± 5.4 | 80.4 ± 8.7 | 63.8 ± 10.9 | 34.4 ± 13.9 | 28.5 ± 10.2 | **78.1 ± 3.8** |
| | | | | Unseen Tasks | | | | |
| 4m | 92.1 ± 3.5 | 28.6 ± 21.6 | 33.0 ± 27.1 | **95.3 ± 3.5** | 48.8 ± 21.1 | 21.6 ± 17.2 | 14.1 ± 5.2 | **61.7 ± 17.7** |
| 5m | 87.1 ± 10.5 | 40.1 ± 25.9 | 33.6 ± 40.2 | **89.1 ± 10.0** | 76.6 ± 14.1 | 77.4 ± 16.0 | 67.2 ± 21.3 | **85.9 ± 11.8** |
| 10m | 90.5 ± 3.8 | 33.9 ± 25.2 | 54.7 ± 44.4 | **93.8 ± 2.2** | 56.2 ± 20.6 | 36.8 ± 20.7 | 32.9 ± 11.3 | **61.3 ± 11.3** |
| 12m | **70.8 ± 15.2** | 10.9 ± 18.9 | 17.2 ± 28.0 | 58.6 ± 11.8 | 24.0 ± 10.5 | 4.0 ± 5.3 | 3.2 ± 3.8 | **35.9 ± 8.1** |
| 7m8m | 18.8 ± 3.1 | 0.8 ± 1.4 | 0.0 ± 0.0 | **25.0 ± 15.1** | 1.6 ± 1.6 | 2.4 ± 2.6 | 0.0 ± 0.0 | **28.1 ± 22.0** |
| 8m9m | 15.8 ± 3.3 | 1.6 ± 1.6 | 0.0 ± 0.0 | **19.6 ± 6.0** | 3.1 ± 3.8 | 3.1 ± 3.1 | 2.3 ± 2.6 | **4.7 ± 2.7** |
| 10m11m | **45.3 ± 11.1** | 0.8 ± 1.4 | 0.0 ± 0.0 | 42.2 ± 7.2 | 19.7 ± 8.9 | 2.4 ± 1.4 | 4.0 ± 3.4 | **29.7 ± 15.4** |
| 10m12m | 1.0 ± 1.5 | 0.0 ± 0.0 | 0.0 ± 0.0 | **1.6 ± 1.6** | 0.0 ± 0.0 | 0.0 ± 0.0 | 0.0 ± 0.0 | **1.6 ± 1.6** |
| 13m15m | 0.0 ± 0.0 | 0.0 ± 0.0 | 0.0 ± 0.0 | **2.3 ± 2.6** | 0.6 ± 1.3 | 0.0 ± 0.0 | 0.0 ± 0.0 | **1.6 ± 1.6** |
| | | | | Medium-Expert | | | | Medium-Replay |
| | | | | Source Tasks | | | | |
| 3m | 67.7 ± 23.7 | 50.1 ± 23.9 | **85.2 ± 17.9** | 73.6 ± 22.0 | 81.1 ± 8.8 | 27.3 ± 25.9 | 41.4 ± 20.1 | **83.6 ± 14.0** |
| 5m6m | **31.3 ± 6.3** | 2.3 ± 2.6 | 1.6 ± 1.6 | 9.4 ± 2.2 | **25.0 ± 3.1** | 0.8 ± 1.4 | 0.8 ± 1.4 | 16.6 ± 4.7 |
| 9m10m | 26.0 ± 13.9 | 27.7 ± 24.1 | 24.3 ± 18.7 | **31.3 ± 14.5** | 33.4 ± 13.1 | 2.3 ± 4.1 | 0.8 ± 1.4 | **34.4 ± 8.0** |
| | | | | Unseen Tasks | | | | |
| 4m | 81.3 ± 18.9 | 41.0 ± 8.0 | 43.9 ± 39.0 | **82.8 ± 13.5** | **61.5 ± 9.0** | 23.4 ± 15.5 | 35.9 ± 12.6 | 55.6 ± 14.5 |
| 5m | 74.0 ± 2.9 | 65.7 ± 10.1 | 33.6 ± 40.2 | **82.8 ± 17.7** | 75.0 ± 24.2 | 54.7 ± 23.5 | 61.7 ± 20.3 | **96.1 ± 4.1** |
| 10m | 78.1 ± 6.7 | 39.8 ± 20.1 | 32.8 ± 38.1 | **82.8 ± 16.8** | 82.4 ± 8.2 | 8.6 ± 8.7 | 11.0 ± 7.8 | **84.4 ± 15.1** |
| 12m | 64.8 ± 24.3 | 9.4 ± 7.9 | 9.4 ± 8.6 | **81.3 ± 20.6** | 83.4 ± 4.5 | 2.3 ± 4.1 | 2.3 ± 2.6 | **84.4 ± 6.6** |
| 7m8m | 13.3 ± 4.5 | 4.0 ± 4.2 | 2.3 ± 4.1 | **15.6 ± 4.4** | 7.3 ± 6.4 | 2.3 ± 2.6 | 1.6 ± 2.7 | **9.4 ± 2.2** |
| 8m9m | 10.2 ± 4.6 | 5.6 ± 4.8 | 9.5 ± 8.6 | **10.9 ± 4.7** | 11.5 ± 3.9 | 0.8 ± 1.4 | 0.8 ± 1.4 | **11.7 ± 8.7** |
| 10m11m | 26.6 ± 4.7 | 8.0 ± 12.2 | 11.8 ± 8.1 | **33.6 ± 8.9** | **46.8 ± 6.6** | 2.3 ± 4.1 | 0.8 ± 1.4 | 35.9 ± 5.2 |
| 10m12m | 0.0 ± 0.0 | 0.0 ± 0.0 | 0.0 ± 0.0 | **1.6 ± 1.6** | 1.6 ± 2.7 | 0.0 ± 0.0 | 0.0 ± 0.0 | **2.3 ± 1.4** |
| 13m15m | 0.8 ± 1.4 | 0.0 ± 0.0 | 0.0 ± 0.0 | **2.3 ± 2.6** | 1.6 ± 1.6 | 0.0 ± 0.0 | 0.0 ± 0.0 | **2.4 ± 1.4** |

seen as an ODIS variant without coordination skill discovery but learning an RL policy to directly select actions. We also implement behavior cloning baselines since they are popular for offline tasks. We introduce a transformer-based behavior cloning baseline (**BC-t**) that has the same structure as ODIS. As decision transformer methods (Chen et al., 2021a) prevail in recent offline literature, we also append return-to-go information to BC-t and thus get a baseline **BC-r**. More details of our implementations and hyper-parameters are reported in Appendix C. We notice that a recent work, MADT (Meng et al., 2021), proposed to train a multi-agent decision transformer with offline training and online tuning. However, we find that MADT is generally not comparable in our settings and put related discussion in Appendix G.

**StarCraft multi-agent micromanagement tasks.** Following previous multi-task MARL methods (Hu et al., 2021; Qin et al., 2022a), we extend the original SMAC maps and sort out three task sets. In each task set, agents will control some units like marines, stalkers, and zealots, but the numbers of controllable agents and target enemies differ from tasks. We refer to our three task sets as marine-easy, marine-hard, and stalker-zealot. Detailed descriptions of these task sets can be found in Appendix A. We adopt the popular QMIX algorithm (Rashid et al., 2020) to collect four classes of offline datasets with different qualities. Following guidelines in single-agent D4RL offline RL benchmarks (Fu et al., 2020; Qin et al., 2022b), the four different dataset qualities are labeled as expert, medium, medium-expert, and medium-replay. We report the detailed properties of these datasets in Appendix B.

We conduct experiments in three task sets with four different data qualities. We train all methods with offline data only from three source tasks and evaluate them in a wide range of unseen tasks. The average test win rates in the task set marine-hard are shown in Table 1, while results of the two other task sets are deferred to Appendix I. The tables report the best test win rates between BC-t and BC-r as BC-best. We find that ODIS generally outperforms other baselines in both source tasks and unseen tasks. ODIS can discover and exploit common coordination skills from multi-task data,

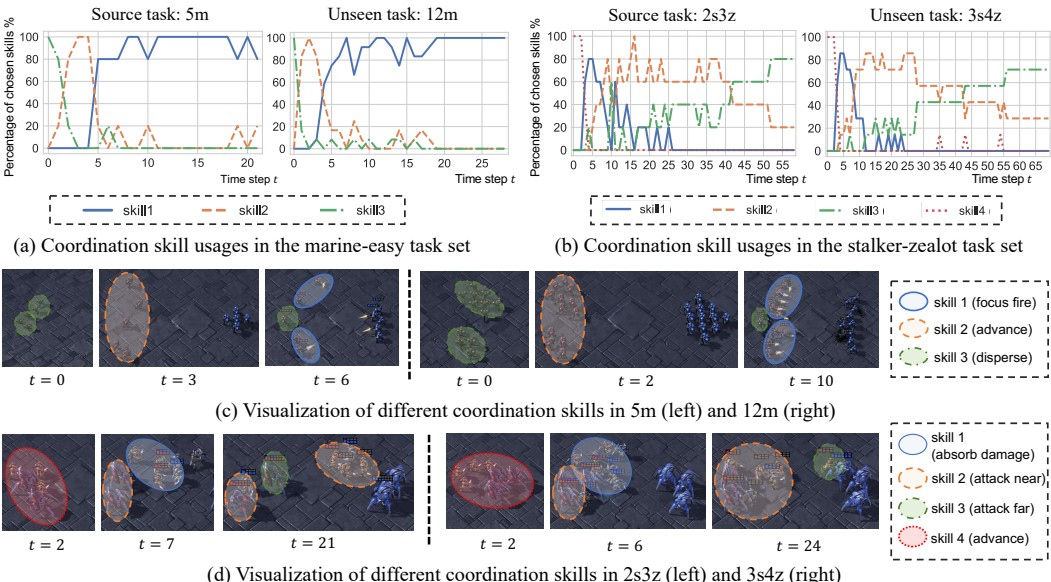

(a) Coordination skill usages in the marine-easy task set

(b) Coordination skill usages in the stalker-zealot task set

(c) Visualization of different coordination skills in 5m (left) and 12m (right)

(d) Visualization of different coordination skills in 2s3z (left) and 3s4z (right)

Figure 3: An illustration of the semantics of discovered coordination skills. We plot the percentages of used coordination skills in two maps from (a) the marine-easy task set and (b) the stalker-zealot task set. We anticipate the semantics of each coordination skill and visualize several test frames from corresponding time steps in (c) 5m and 12m maps and (d) 2s3z and 3s4z maps. Agents in colored circle choose the corresponding coordination skill with the same color in legends of (a) and (b).

resulting in superior and stable performance compared with UPDeT-l and UPDeT-m, which cannot generalize well among different levels of tasks. We notice that behavior cloning methods present comparable performance, especially in expert datasets, indicating that our proposed transformer structure also enhances generalization across tasks. However, these behavior cloning baselines cannot effectively exploit multi-task data to further improve its performance, although we add additional global return-to-go information in BC-r.

To evaluate the validity of ODIS, we also compare ODIS with other single-task MARL approaches that can perform offline training, including ICQ (Yang et al., 2021) and QPLEX (Wang et al., 2021a). These works can adopt offline learning manners but cannot directly learn from multi-task data. We find that ODIS can exhibit comparable performance with these single-task offline MARL methods even in the unseen task of ODIS. These experiments reveal that the multi-task offline training manner of ODIS can indeed promote generalization to different tasks. We defer the experimental results and detailed discussion to Appendix D.

**Cooperative navigation tasks.** To further validate the effectiveness of ODIS, we also conduct experiments in several cooperative navigation tasks from the multi-agent particle environment (Lowe et al., 2017), where varying numbers of agents learn to reach corresponding landmarks. We notice that ODIS still shows superior performance in both source and unseen tasks. The results and detailed descriptions of these tasks can be found in Appendix F.

## 5.2 Semantics of Discovered Coordination Skills

To investigate how the discovered coordination skill helps decision-making, we deploy ODIS agents to different tasks and record chosen coordination skills in test episodes. As shown in Figure 3, we exhibit the percentage of each coordination skill usage from policies learned in marine-easy and stalker-zealot task sets with expert datasets. For the marine-easy task set in Figure 3(a), we adopt a coordination skill number of three and find that the curves of each coordination skill are pretty similar between a source task 5m and an unseen task 12m despite different episode lengths. Further visualization shows that the three coordination skills represent different coordinative behaviors. The agents learn to utilize skill 3 to disperse in different directions and then use skill 2 to advance. When enemies are in the attack range, most agents choose skill 1 to focus fire on specific enemies, and a few

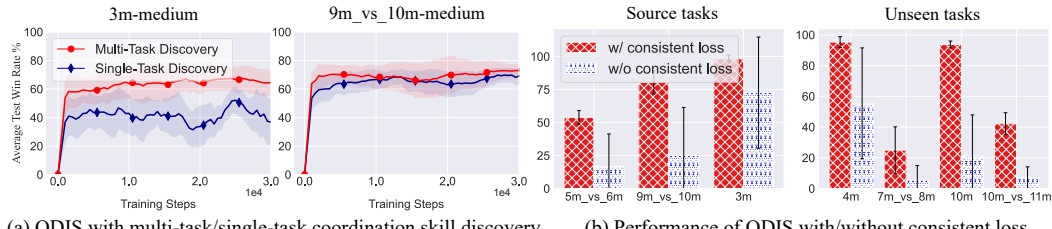

(a) ODIS with multi-task/single-task coordination skill discovery    (b) Performance of ODIS with/without consistent loss

Figure 4: (a) average test win rates of ODIS when performing multi-task/single-task coordination skill discovery. (b) average test win rates of ODIS with/without consistent loss.

agents choose skill 2 and skill 3 to pursue enemies or run away from fire. For the stalker-zealot task set in Figure 3(b), we adopt a coordination skill number of four as the task is a little more complicated. We exhibit the coordination skill usages in a source task 2s3z and an unseen task 3s4z, whose curves are also similar. The visualization indicates that the chosen coordination skill may present different semantics from marine-easy. In the beginning, agents use skill 4 to advance. When approaching the enemies, some agents choose to attack nearby enemies with skill 2, and others learn to absorb damage forwardly with skill 1. We find that the stalker, a melee unit, is more willing to draw enemy fire than the ranging unit stalker. After that, a few agents will use skill 3 to attack enemies far away, and more agents tend to choose skill 3 when nearby enemies fall.

## 5.3 ABLATION STUDIES

In ablation studies, we investigate the effectiveness of each component in our proposed ODIS structure. First, we try to find whether discovering coordination skills from multiple tasks can yield better performance in a particular task. We perform ODIS offline training separately with multi-task and single-task coordination skill discovery. ODIS with multi-task coordination skill discovery can utilize data from the two other tasks to discover potentially more effective coordination skills, while ODIS with single-task coordination skill discovery can only access data from the corresponding task. In coordination policy learning, both two approaches are learned from single-task data. Results in Figure 4(a) show that ODIS with multi-task coordination skill discovery acquires better performance, indicating that shared coordination skills across tasks can benefit other tasks.

We also conduct experiments to investigate our proposed consistent loss, which helps learn effective local representations for choosing coordination skills from a coordinative perspective as the state encoder does. We run experiments with two variants of ODIS with and without consistent loss in the marine-hard task set and present average test win rates in three source tasks and four unseen tasks. As shown in Figure 4(b), the performance of ODIS with consistent loss is significantly better than ODIS without consistent loss, which shows the proposed loss can maintain the consistency of coordination skills chosen between coordination skill discovery and coordination policy learning.

Furthermore, the number of coordination skills is a key hyper-parameter of ODIS, so we conduct experiments to compare performances with different coordination skill numbers in Table 7. We find the performance of ODIS is not sensitive with coordination skills numbers ranging from 3 to 16. However, adopting less skill numbers or random selection will significantly fail, as discussed in Appendix E.

## 6 CONCLUSION

We propose ODIS, an offline MARL algorithm applying coordination skill discovery from multi-task data, realizing multi-task generalization in a fully data-driven manner. ODIS extracts and utilizes coordination skills shared among different tasks and thus acquires superior performance in both source and unseen tasks. The effectiveness of ODIS indicates that underlying coordination skills from multi-task data can be crucial for generalization in cooperative MARL. As the discrete coordination skill might be limited when facing dissimilar tasks, developing general representations among dissimilar tasks from multi-task or many-task data is a promising direction in the future. In addition, research on offline MARL algorithms and benchmarks in various domains will also be helpful to real-world applications of MARL.

## ACKNOWLEDGMENTS

This work is supported by National Key Research and Development Program of China (2020AAA0107200), the National Science Foundation of China (61921006, 62276126), and the Natural Science Foundation of Jiangsu (BK20221442). We thank Feng Chen, Xinyu Yang, and the anonymous reviewers for their support and helpful discussions on improving the paper.

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

## A DESCRIPTIONS OF DEFINED TASK SETS IN SMAC

The StarCraft multi-agent challenge (SMAC) (Samvelyan et al., 2019) is a widely used cooperative multi-agent environment containing different types of StarCraft micromanagement tasks. In this paper, we sort out three task sets and call them marine-easy, marine-hard, and stalker-zealot, respectively. The marine-easy and marine-hard task sets include several marine battle tasks where several ally marines need to beat the same number or a larger number of enemy marines. In the marine-easy task set, the number of enemy marines equals the number of ally marines, while tasks from the marine-hard task set contain both equal numbers and larger numbers of enemy marines. The stalker-zealot task set includes several tasks with symmetric stalkers and zealots on each side. We exhibit illustrations of these tasks in Figure 5(a) and Figure 5(b). For our goal of generalization to unseen tasks with limited sources, we select three tasks from the task set as training tasks, and the other tasks are only for evaluation. The detailed properties of these task sets can be seen in Tables 2, 3, and 4, respectively.

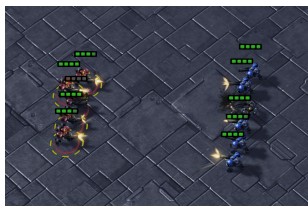

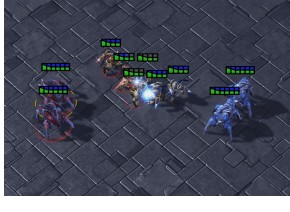

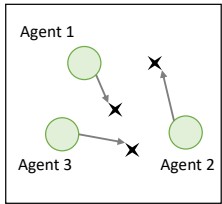

| (a) 5m_vs_6m (SMAC) | (b) 2s3z (SMAC) | (c) Cooperative navigation (3 agents) |

Figure 5: Illustrations of different kinds of tasks in the experiments section. (a) The marine battle task 5m_vs_6m of SMAC, where agents control 5 controllable marines to beat 6 built-in-AI marines. (b) The heterogeneous task 2s3z of SMAC, where agents control 2 stalkers and 3 zealots to beat the same number of built-in-AI units. (c) The cooperative navigation task with 3 agents from the multi-agent particle environment, where 3 agents need to reach corresponding landmarks.

Table 2: Descriptions of tasks in marine-easy task set.

| Task type | Task | Ally units | Enemy units | Properties |
|---|---|---|---|---|
| Source tasks | 3m | 3 Marines | 3 Marines | homogeneous & symmetric |
| | 5m | 5 Marines | 5 Marines | homogeneous & symmetric |
| | 10m | 10 Marines | 10 Marines | homogeneous & symmetric |
| Unseen tasks | 4m | 4 Marines | 4 Marines | homogeneous & symmetric |
| | 6m | 6 Marines | 6 Marines | homogeneous & symmetric |
| | 7m | 7 Marines | 7 Marines | homogeneous & symmetric |
| | 8m | 8 Marines | 8 Marines | homogeneous & symmetric |
| | 9m | 9 Marines | 9 Marines | homogeneous & symmetric |
| | 11m | 11 Marines | 11 Marines | homogeneous & symmetric |
| | 12m | 12 Marines | 12 Marines | homogeneous & symmetric |

## B PROPERTIES OF OFFLINE DATASETS

As stated in the experiments section, we construct offline datasets based on the PyMARL implementation[2] of the MARL algorithm QMIX (Rashid et al., 2020). Following the popular single-agent offline reinforcement learning benchmark D4RL (Fu et al., 2020), we collect data with four types of qualities called expert, medium, medium-expert, and medium-replay, respectively. Definitions of these four qualities are listed below:

- The **expert** dataset contains trajectory data collected by a QMIX policy trained with $2,000,000$ steps of environment interactions. We also record the test win rate of the trained QMIX policy (as the expert policy) for constructing medium datasets.

[2] https://github.com/oxwhirl/pymarl

Table 3: Descriptions of tasks in marine-hard task set.

| Task type | Task | Ally units | Enemy units | Properties |
|---|---|---|---|---|
| Source tasks | 3m | 3 Marines | 3 Marines | homogeneous & symmetric |
| | 5m_vs_6m | 5 Marines | 6 Marines | homogeneous & asymmetric |
| | 9m_vs_10m | 9 Marines | 10 Marines | homogeneous & asymmetric |
| Unseen tasks | 4m | 4 Marines | 4 Marines | homogeneous & symmetric |
| | 5m | 5 Marines | 5 Marines | homogeneous & symmetric |
| | 10m | 10 Marines | 10 Marines | homogeneous & symmetric |
| | 12m | 12 Marines | 12 Marines | homogeneous & symmetric |
| | 7m_vs_8m | 7 Marines | 8 Marines | homogeneous & asymmetric |
| | 8m_vs_9m | 8 Marines | 9 Marines | homogeneous & asymmetric |
| | 10m_vs_11m | 10 Marines | 11 Marines | homogeneous & asymmetric |
| | 10m_vs_12m | 10 Marines | 12 Marines | homogeneous & asymmetric |
| | 13m_vs_15m | 13 Marines | 15 Marines | homogeneous & asymmetric |

Table 4: Descriptions of tasks in stalker-zealot task set.

| Task type | Task | Ally units | Enemy units | Properties |
|---|---|---|---|---|
| Source tasks | 2s3z | 2 Stalkers, 3 Zealots | 2 Stalkers, 3 Zealots | heterogeneous & symmetric |
| | 2s4z | 2 Stalkers, 4 Zealots | 2 Stalkers, 4 Zealots | heterogeneous & symmetric |
| | 3s5z | 3 Stalkers, 5 Zealots | 3 Stalkers, 5 Zealots | heterogeneous & symmetric |
| Unseen tasks | 1s3z | 1 Stalkers, 3 Zealots | 1 Stalkers, 3 Zealots | heterogeneous & symmetric |
| | 1s4z | 1 Stalkers, 4 Zealots | 1 Stalkers, 4 Zealots | heterogeneous & symmetric |
| | 1s5z | 1 Stalkers, 5 Zealots | 1 Stalkers, 5 Zealots | heterogeneous & symmetric |
| | 2s5z | 2 Stalkers, 5 Zealots | 2 Stalkers, 5 Zealots | heterogeneous & symmetric |
| | 3s3z | 3 Stalkers, 3 Zealots | 3 Stalkers, 3 Zealots | heterogeneous & symmetric |
| | 3s4z | 3 Stalkers, 4 Zealots | 3 Stalkers, 4 Zealots | heterogeneous & symmetric |
| | 4s3z | 4 Stalkers, 3 Zealots | 4 Stalkers, 3 Zealots | heterogeneous & symmetric |
| | 4s4z | 4 Stalkers, 4 Zealots | 4 Stalkers, 4 Zealots | heterogeneous & symmetric |
| | 4s5z | 4 Stalkers, 5 Zealots | 4 Stalkers, 5 Zealots | heterogeneous & symmetric |

- The **medium** dataset contains trajectory data collected by a QMIX policy (as the medium policy) whose test win rate is half of the expert QMIX policy.

- The **medium-expert** dataset mixes data from the expert dataset and the medium dataset to acquire a more diverse dataset.

- The **medium-replay** dataset is the replay buffer of the medium policy, containing trajectory data with lower qualities.

As we consider generalization to unseen tasks, we only require offline datasets in the source tasks of the three task sets mentioned above. For the expert and medium datasets, we collect $2,000$ trajectories to construct each dataset, and thus the medium-expert dataset contains $4,000$ trajectories as a mixture of the expert and medium data. The trajectory number of the medium-replay dataset depends on

Table 5: Properties of offline datasets with different qualities.

| Task | Quality | # Trajectories | Average return | Average win rate |
|------|---------|----------------|----------------|------------------|
| 3m | expert | 2000 | 19.8929 | 0.9910 |
| | medium | 2000 | 13.9869 | 0.5402 |
| | medium-expert | 4000 | 16.9399 | 0.7656 |
| | medium-replay | 3630 | N/A | N/A |
| 5m | expert | 2000 | 19.9380 | 0.9937 |
| | medium | 2000 | 17.3288 | 0.7411 |
| | medium-expert | 4000 | 18.6334 | 0.8674 |
| | medium-replay | 771 | N/A | N/A |
| 10m | expert | 2000 | 19.9438 | 0.9922 |
| | medium | 2000 | 16.6297 | 0.5413 |
| | medium-expert | 4000 | 18.2595 | 0.7626 |
| | medium-replay | 571 | N/A | N/A |
| 5m_vs_6m | expert | 2000 | 17.3424 | 0.7185 |
| | medium | 2000 | 12.6408 | 0.2751 |
| | medium-expert | 4000 | 14.9916 | 0.4968 |
| | medium-replay | 32607 | N/A | N/A |
| 9m_vs_10m | expert | 2000 | 19.6140 | 0.9431 |
| | medium | 2000 | 15.5049 | 0.4146 |
| | medium-expert | 4000 | 17.5594 | 0.6789 |
| | medium-replay | 13731 | N/A | N/A |
| 2s3z | expert | 2000 | 19.7655 | 0.9602 |
| | medium | 2000 | 16.6279 | 0.4465 |
| | medium-expert | 4000 | 18.1967 | 0.7034 |
| | medium-replay | 4505 | N/A | N/A |
| 2s4z | expert | 2000 | 19.7402 | 0.9509 |
| | medium | 2000 | 16.8735 | 0.4965 |
| | medium-expert | 4000 | 18.3069 | 0.7237 |
| | medium-replay | 6172 | N/A | N/A |
| 3s5z | expert | 2000 | 19.7850 | 0.9518 |
| | medium | 2000 | 16.3126 | 0.3114 |
| | medium-expert | 4000 | 18.0488 | 0.6316 |
| | medium-replay | 11528 | N/A | N/A |

the number of sampling trajectories before the medium policy stops training. When performing multi-task offline training in our experiments, we select up to $2,000$ trajectories of data from each source task. When the medium-replay dataset contains less than $2,000$ trajectories of data, we select all trajectories. Data from different tasks are merged into a multi-task dataset to ensure that the policy is trained with multi-task data simultaneously.

## C  STRUCTURES, HYPER-PARAMETERS, AND TRAINING DETAILS OF ODIS

In this section, we will exhibit the network structure, hyper-parameter choices, and other training details of ODIS. As illustrated in Figure 6, the network structure of ODIS mainly contains four components, the action decoder, state encoder, observation encoder, and mixing network, respectively. These four components apply the attention mechanism to process alterable state and observation spaces. The observation encoder and the action decoder take the observation as input, and the state encoder and the mixing network take the global state as input.

As presented in the preliminaries, we decompose the observation information $o_i$ of an agent $i$ into several portions including its own/environmental information $o_i^{\mathrm{own}}$ and other entities' information $\{o_{i,j}^{\mathrm{other}}\}$. Each kind of portion is fed into a separate fully connected layer to acquire an embedding

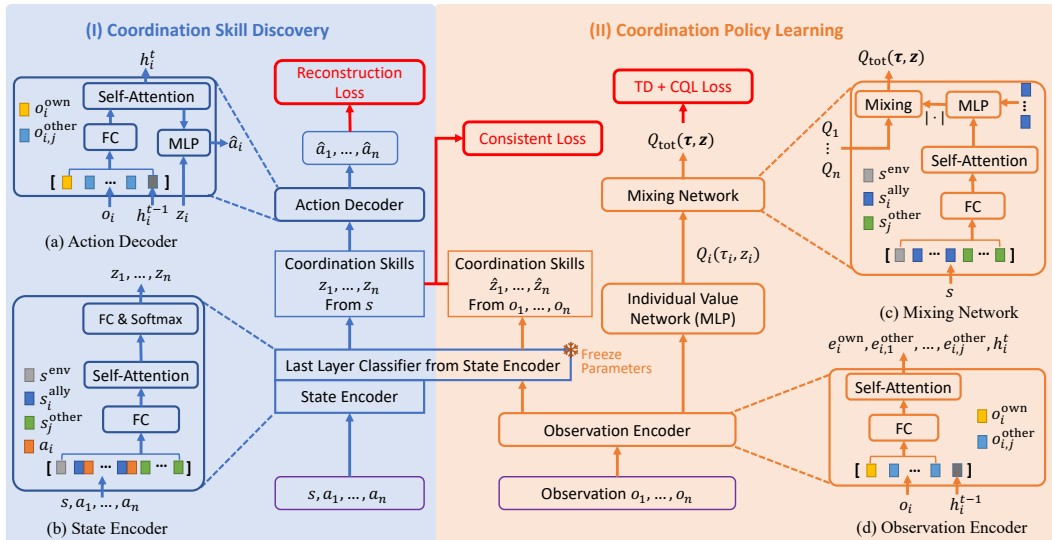

Figure 6: The network structure of ODIS. In coordination skill discovery, ODIS leverages (a) a local action decoder and (b) a global state encoder to discover coordination skills from multi-task offline data. In coordination policy learning, ODIS performs MARL with the CTDE paradigm to train individual coordination policies with the assistance of (c) a mixing network. The individual coordination policy contains (d) an observation encoder to extract representations from local information.

Table 6: Hyper-parameters of ODIS.

| Hyper-parameter | Value |
| --- | --- |
| hidden layer dimension | 64 |
| attention dimension | 64 |
| $\alpha$ | 5.0 |
| $\beta$ | 0.001 |
| $\lambda$ | 5.0 |
| coordination skill number | 3 (marine-easy); 5 (marine-hard); 4 (stalker-zealot) |
| steps of coordination skill discovery | 15000 |
| optimizer | Adam |
| learning rate | 0.0005 |

with the dimension of $64$. For the self-attention network, we generate $Q, K, V$ with fully connected layers from the embedding sequence and then perform self-attention (Vaswani et al., 2017) with the input and output dimensions of $64$. The output sequence can be formalized as the representation of the agent $i$'s own information $e_i^{\mathrm{own}}$ and the representation to other entities $\{e_{i,j}^{\mathrm{other}}\}$ in agent $i$'s view. To handle the partial observability, we append the input sequence with a historical embedding $h_i^{t-1}$ when applying self-attention and thus acquire the output of $h_i^t$, following the UPDeT structure (Hu et al., 2021). For the observation encoder, we only select the own information representation $e_i^{\mathrm{own}}$ to calculate Q-values and feed it into the individual network or the coordination skill classifier. For the action decoder, we embed the chosen skill to a vector of $64$ dimension and concatenate it with the output sequence. When predicting actions, we divide available actions into interactive actions and non-interactive actions, where the interactive action means an action that needs to interact with an entity and non-interactive action means an action that is only relevant to an agent itself. We output the values of non-interactive actions from $e_i^{\mathrm{own}}$ and the values of interactive actions with entity $j$ from corresponding $e_{i,j}^{\mathrm{other}}$ and further concatenate them to select the action with the maximal value.

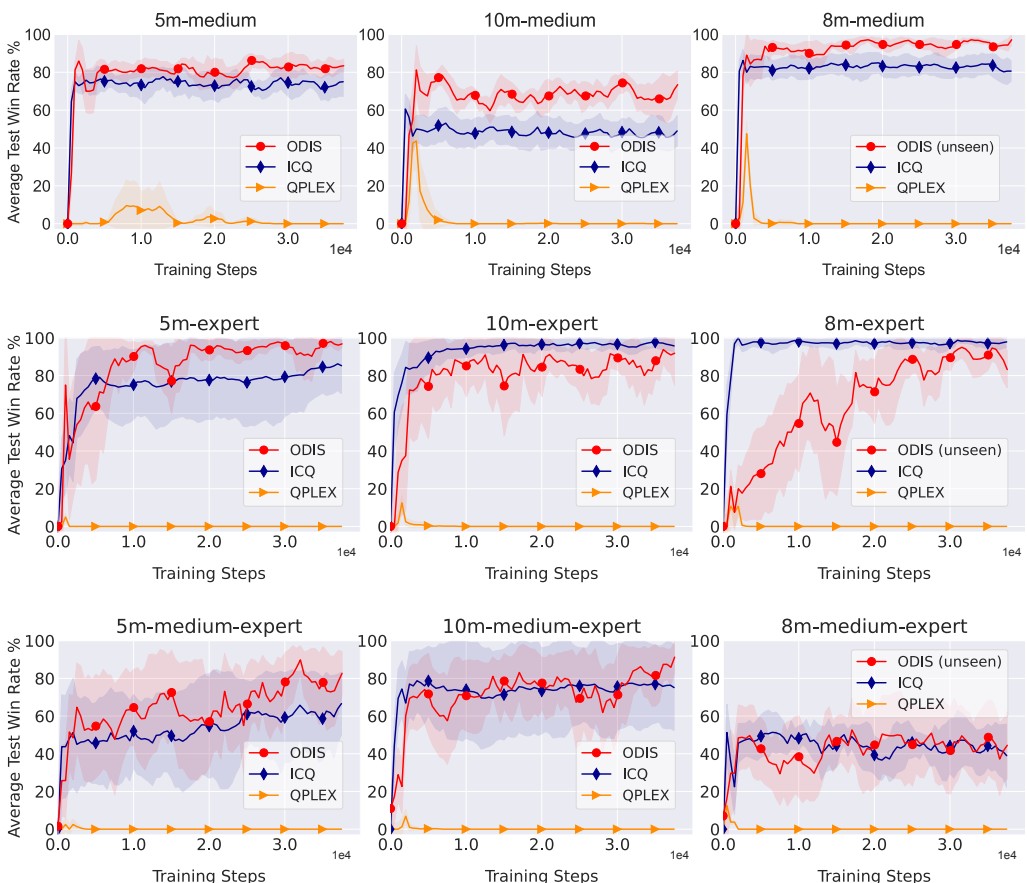

Figure 7: Average test win rates of ODIS, ICQ, and QPLEX in three maps of the marine-easy task set with medium datasets (top), medium datasets (middle), and medium-expert datasets (bottom). Here ODIS is trained with offline data from three source tasks, 3m, 5m, and 10m, as shown in Table 2, while ICQ and QPLEX are trained with corresponding single-task data. Note that 8m is an *unseen* task for ODIS, where ODIS can still acquire comparable performance with the two other offline MARL algorithms using offline data of 8m for training.

We also decompose the state information $s$ into several portions including the environment information $s^{\mathrm{env}}$, ally information $\{s_i^{\mathrm{ally}}\}$, and other entities' information $\{s_j^{\mathrm{other}}\}$. Like the process of the observation, we feed them into a separate fully connected layer to acquire embeddings with the dimension of $64$ and further calculate Q, K, V and perform self-attention. The dimensions of Q and K are set to $8$ to reduce computation as the state may contain numerous entities, while V remains a dimension of $64$. The output sequence consists of the representation of each portion. For the mixing network, we utilize all ally information to generate non-negative weights by calculating the absolute value of MLP outputs. For the state encoder, we additionally append the action $a_i$ to the corresponding ally information $s_i^{\mathrm{ally}}$ to perform self-attention and use ally representations from the output sequence as the input of the coordination skill classifier. The coordination skill classifier contains a fully connected layer along with the softmax function to compute the probability of each coordination skill. The individual value network also contains a fully connected layer to transform local representations into Q-values.

Besides the above network structure, ODIS needs two training phases to perform coordination skill discovery and coordination policy learning separately. We implement ODIS with the aforementioned PyMARL framework to ensure that the mechanics irrelevant to the algorithm are the same as previous methods. Other specific hyper-parameters are listed in Table 6. All the tabular results show the performance of ODIS with $50,000$ optimization steps, and the steps of the coordination policy

Table 7: Average test win rates of the final policies trained with different coordination skill numbers (abbreviated as "skill num.") in the task set marine-hard with medium data quality. The listed performance is averaged over five random seeds. We abbreviate asymmetric task names for simplicity. For example, the task name "5m6m" denotes the SMAC map "5m_vs_6m". The column name "rand. skill" stands for random coordination skill selection.

| Task | skill num. 1 | skill num. 3 | skill num. 5 | skill num. 8 | skill num. 16 | rand. skill |
|---|---|---|---|---|---|---|
| | | | Source tasks | | | |
| 3m | $67.8 \pm 12.1$ | $72.7 \pm 10.2$ | $\mathbf{85.9 \pm 10.5}$ | $85.2 \pm 8.9$ | $80.5 \pm 5.6$ | $0.0 \pm 0.0$ |
| 5m6m | $21.4 \pm 5.5$ | $\mathbf{22.7 \pm 10.5}$ | $22.7 \pm 7.1$ | $14.8 \pm 11.8$ | $18.8 \pm 6.2$ | $0.0 \pm 0.0$ |
| 9m10m | $66.1 \pm 9.2$ | $47.7 \pm 8.1$ | $\mathbf{78.1 \pm 3.8}$ | $45.3 \pm 12.2$ | $65.6 \pm 12.7$ | $0.0 \pm 0.0$ |
| | | | Unseen Tasks | | | |
| 4m | $44.3 \pm 12.5$ | $\mathbf{78.1 \pm 9.4}$ | $61.7 \pm 17.7$ | $65.6 \pm 5.8$ | $64.1 \pm 10.9$ | $0.0 \pm 0.0$ |
| 5m | $73.2 \pm 12.4$ | $57.0 \pm 24.5$ | $85.9 \pm 11.8$ | $78.1 \pm 4.9$ | $\mathbf{100.0 \pm 0.0}$ | $0.0 \pm 0.0$ |
| 10m | $37.1 \pm 23.4$ | $55.5 \pm 30.7$ | $61.3 \pm 11.3$ | $53.9 \pm 7.1$ | $\mathbf{62.5 \pm 36.0}$ | $0.0 \pm 0.0$ |
| 12m | $22.7 \pm 26.8$ | $32.0 \pm 37.8$ | $\mathbf{35.9 \pm 8.1}$ | $15.6 \pm 13.1$ | $30.5 \pm 25.4$ | $0.0 \pm 0.0$ |
| 7m8m | $1.6 \pm 1.6$ | $3.1 \pm 5.4$ | $\mathbf{28.1 \pm 22.0}$ | $1.6 \pm 1.6$ | $4.7 \pm 5.2$ | $0.0 \pm 0.0$ |
| 8m9m | $3.1 \pm 3.8$ | $1.6 \pm 1.6$ | $\mathbf{4.7 \pm 2.7}$ | $1.6 \pm 1.6$ | $3.1 \pm 3.1$ | $0.0 \pm 0.0$ |
| 10m11m | $21.2 \pm 6.7$ | $21.1 \pm 13.5$ | $\mathbf{29.7 \pm 15.4}$ | $3.1 \pm 5.4$ | $10.2 \pm 8.1$ | $0.0 \pm 0.0$ |
| 10m12m | $0.0 \pm 0.0$ | $0.0 \pm 0.0$ | $\mathbf{1.6 \pm 1.6}$ | $0.0 \pm 0.0$ | $0.0 \pm 0.0$ | $0.0 \pm 0.0$ |
| 13m15m | $0.0 \pm 0.0$ | $0.0 \pm 0.0$ | $\mathbf{1.6 \pm 1.6}$ | $0.0 \pm 0.0$ | $0.0 \pm 0.0$ | $0.0 \pm 0.0$ |

Table 8: Properties of offline datasets with different qualities in cooperative navigation tasks.

| Task | Quality | # Trajectories | Average return | Average success rate |
|---|---|---|---|---|
| CN-2 | expert | 2000 | 1.0000 | 1.0000 |
| | medium | 2000 | 0.6152 | 0.6152 |
| CN-4 | expert | 2000 | 0.7173 | 0.7173 |
| | medium | 2000 | 0.4273 | 0.4273 |

learning phase are the subtraction of the coordination skill discovery steps from the total steps. The training process of ODIS with an NVIDIA GeForce RTX 2080Ti GPU and a 32-core CPU costs 12-14 hours typically. Our released implementation of ODIS, along with the provided offline datasets, follows Apache License 2.0, the same as the PyMARL framework.

# D  COMPARISONS WITH SINGLE-TASK OFFLINE MARL METHODS

Although ODIS aims at the multi-task offline MARL domain, we also compare ODIS with offline MARL methods trained in single-task in the marine-easy task set, where ODIS utilizes three source tasks for training. We select two baselines, ICQ and QPLEX, for comparison. ICQ (Yang et al., 2021) is a conservative-style offline MARL algorithm that can handle the severe distribution shift issue in offline MARL. QPLEX (Wang et al., 2021a) claims the IGM-complete value decomposition can benefit offline training. As the two baselines cannot leverage multi-task offline data, we feed the corresponding task dataset into these algorithms to train policies independently. We evaluate the performance in three tasks, including 5m, 10m, and 8m, where 8m is an unseen task for ODIS. To train the two baselines in task 8m, we also collect data with a QMIX policy like previous statements, and the properties of the data are exhibited in Table 5. The data in 8m is only used for the two baselines but remains unseen to ODIS. We conduct experiments with medium, expert, and medium-expert data qualities. As shown in Figure 7, ODIS outperforms other baselines in both source tasks 5m and 8m and the unseen task 8m, indicating that with learning through multi-task data, ODIS can not only perform better than other single-task offline MARL methods in most tasks but present tremendous zero-shot task generalization to unseen tasks. ICQ can generally acquire a good performance in expert data, but its conservative paradigm may limit the performance. QPLEX struggles in these datasets without particular tuning, and we speculate that it is because our datasets cannot provide

Table 9: Average test success rates in the cooperative navigation task set with different data qualities.

| | Expert | | | | |
|---|---|---|---|---|---|
| Task | ODIS (ours) | BC-best | UPDeT-m | UPDeT-l | QMIX (online) |
| | Source tasks | | | | |
| CN-2 | **100.0 ± 0.0** | 99.0 ± 1.5 | 90.6 ± 6.8 | 91.7 ± 9.7 | 100.0 |
| CN-4 | **46.2 ± 13.6** | 43.8 ± 7.7 | 15.6 ± 9.2 | 5.2 ± 5.3 | 71.7 |
| | Unseen tasks | | | | |
| CN-3 | **85.6 ± 7.6** | 72.9 ± 2.9 | 47.9 ± 10.3 | 30.2 ± 14.5 | 83.4 |
| CN-5 | **20.0 ± 7.8** | 12.5 ± 6.8 | 2.1 ± 2.9 | 0.0 ± 0.0 | 0.0 |
| | Medium | | | | |
| Task | ODIS (ours) | BC-t | UPDeT-m | UPDeT-l | QMIX (online) |
| | Source tasks | | | | |
| CN-2 | **65.0 ± 5.4** | 50.0 ± 5.1 | 35.4 ± 12.1 | 47.9 ± 9.7 | 100.0 |
| CN-4 | **28.7 ± 6.7** | 24.0 ± 3.9 | 4.2 ± 2.9 | 8.3 ± 2.9 | 71.7 |
| | Unseen tasks | | | | |
| CN-3 | **43.8 ± 5.2** | 43.8 ± 2.6 | 14.6 ± 3.9 | 26.0 ± 5.9 | 83.4 |
| CN-5 | **8.1 ± 2.5** | 7.3 ± 3.9 | 0.0 ± 0.0 | 1.0 ± 1.5 | 0.0 |

diverse data for QPLEX to perform policy exploitation and lead to large extrapolation errors. The empirical results show that discovering and sharing coordination skills can be efficient and powerful across different tasks.

## E EXPERIMENTS WITH DIFFERENT COORDINATION SKILL NUMBERS

The size of the coordination skill set $\mathcal{Z}$ is a key hyper-parameter in ODIS, which represents the number of actions that can be chosen in the coordination policy. We conduct experiments in the marine-hard task-set with medium quality to investigate whether decision-making in the coordination skill space works. We select the coordination skill number of 5 as the default setting for our main experiments. As shown in Table 7, we can find that the choices of coordination skill numbers 3, 5, 8, and 16 exhibit comparable performances in most unseen tasks, indicating that ODIS does not need a sophisticated tuning at the coordination skill number. We finally choose the coordination skill number of 5 in this task set because it obtains a generally better performance. On the other hand, we can find that a coordination skill number of 1 cannot generalize to most unseen tasks, as its performance entirely depends on the reconstructive ability of the action decoder with only 1 coordination skill that can be selected. We also evaluate the performance of ODIS with randomly choosing a coordination skill when the total coordination skill number is 5, where the coordination policy is not trained and we can only rely on the action decoder for decision-making. The results of ODIS with random skills exhibit all average test win rates of 0, indicating that the discovered coordination skill can be sufficiently utilized by the action decoder to generate task-relevant actions and cannot be simply disentangled from the framework.

## F RESULTS ON COOPERATIVE NAVIGATION TASKS

To further evaluate the effectiveness of ODIS, we design and conduct experiments in a task set from the cooperative navigation environment. Cooperative navigation is a series of tasks from the multi-agent particle environment (MPE) raised by MADDPG (Lowe et al., 2017). In this environment, a number of agents try to reach corresponding landmarks and only acquire a reward of 1 when they all successfully reach the landmarks. A visualization of this task can be found in Figure 5(c). We add discrete control support to the original cooperative navigation tasks, where agents can execute actions of moving towards four directions and a "none" operation. We design a task set in the cooperative

Table 10: Average test win rates of ODIS and MADT in the marine-hard task set with expert and medium data qualities.

| Expert | | | Medium | | |
|---|---|---|---|---|---|
| Task | ODIS (ours) | MADT | Task | ODIS (ours) | MADT |
| Source tasks | | | | | |
| 3m | $\mathbf{98.4 \pm 2.7}$ | $88.5 \pm 3.9$ | 3m | $\mathbf{85.9 \pm 10.5}$ | $4.2 \pm 1.5$ |
| 5m_vs_6m | $\mathbf{53.9 \pm 5.1}$ | $3.1 \pm 0.0$ | 5m_vs_6m | $\mathbf{22.7 \pm 7.1}$ | $21.9 \pm 2.6$ |
| 9m_vs_10m | $\mathbf{80.4 \pm 8.7}$ | $1.0 \pm 1.5$ | 9m_vs_10m | $\mathbf{78.1 \pm 3.8}$ | $14.6 \pm 7.4$ |
| Unseen tasks | | | | | |
| 4m. | $\mathbf{95.3 \pm 3.5}$ | $83.3 \pm 5.3$ | 4m | $\mathbf{61.7 \pm 17.7}$ | $31.2 \pm 15.9$ |
| 5m | $\mathbf{89.1 \pm 10.0}$ | $75.0 \pm 6.8$ | 5m | $\mathbf{85.9 \pm 11.8}$ | $63.5 \pm 12.6$ |
| 10m | $\mathbf{93.8 \pm 2.2}$ | $1.0 \pm 1.5$ | 10m | $\mathbf{61.3 \pm 11.3}$ | $33.3 \pm 14.1$ |
| 12m | $\mathbf{58.6 \pm 11.8}$ | $0.0 \pm 0.0$ | 12m | $\mathbf{35.9 \pm 8.1}$ | $0.0 \pm 0.0$ |
| 7m_vs_8m | $\mathbf{25.0 \pm 15.1}$ | $1.0 \pm 1.5$ | 7m_vs_8m | $\mathbf{28.1 \pm 22.0}$ | $1.0 \pm 1.5$ |
| 8m_vs_9m | $\mathbf{19.6 \pm 6.0}$ | $0.0 \pm 0.0$ | 8m_vs_9m | $\mathbf{4.7 \pm 2.7}$ | $0.0 \pm 0.0$ |
| 10m_vs_11m | $\mathbf{42.2 \pm 7.2}$ | $0.0 \pm 0.0$ | 10m_vs_11m | $\mathbf{29.7 \pm 15.4}$ | $0.0 \pm 0.0$ |
| 10m_vs_12m | $\mathbf{1.6 \pm 1.6}$ | $0.0 \pm 0.0$ | 10m_vs_12m | $\mathbf{1.6 \pm 1.6}$ | $0.0 \pm 0.0$ |
| 13m_vs_15m | $\mathbf{2.3 \pm 2.6}$ | $0.0 \pm 0.0$ | 13m_vs_15m | $\mathbf{1.6 \pm 1.6}$ | $0.0 \pm 0.0$ |

Table 11: Average test win rates of ODIS and MADT in a small task set (including source tasks of 3m and 5m, and unseen tasks of 4m and 6m) with medium data qualities.

| Source Task | ODIS (ours) | MADT | Unseen Task | ODIS (ours) | MADT |
|---|---|---|---|---|---|
| 3m | $\mathbf{85.9 \pm 10.5}$ | $60.4 \pm 10.3$ | 4m | $\mathbf{95.3 \pm 3.5}$ | $51.0 \pm 5.3$ |
| 5m | $\mathbf{85.9 \pm 11.8}$ | $74.0 \pm 3.9$ | 6m | $\mathbf{78.0 \pm 9.0}$ | $10.4 \pm 7.4$ |

navigation environment by setting different numbers of agents and naming the cooperative navigation task with $n$ agents CN-$n$. Thus we set up four tasks named CN-2, CN-3, CN-4, and CN-5, respectively. We collect different qualities of data using the QMIX algorithm. The detailed dataset properties are shown in Table 8.

We compare ODIS with the aforementioned baselines, including BC-t, BC-r, UPDeT-m, and UPDeT-l, and evaluate in the expert and medium datasets, respectively. The results are exhibited in Table 9. ODIS outperforms other baselines in both expert data and medium data. As a reference, we also put the results of online QMIX algorithms in the table. We find that in two unseen tasks, CN-3 and CN-5, ODIS trained with expert data can acquire better performances than a learn-from-scratch QMIX algorithm. A notable observation is that ODIS can generalize to CN-5, while an online QMIX will fail to learn a valid policy in this task.

## G  EVALUATIONS OF ODIS AND MADT

MADT (Meng et al., 2021) is a recent approach to training a multi-agent decision transformer with offline training and optional online tuning. However, MADT is not a baseline originally designed for simultaneous multi-task training. We compare ODIS with MADT in the marine-hard task set with expert and medium data qualities, and exhibit the results in Table 10. We find MADT cannot generally learn a valid policy in our multi-task settings, and we provide the following three explanations:

1. MADT utilizes feature encoding and action masking techniques to deal with different input shapes. All inputs are encoded into the same shape with zero padding, and the action space has to be aligned to the maximal action space with unavailable actions masked. These techniques may induce poor generalization when the input shape changes dramatically (e.g., the observation size of task 3m is $42$ while for 10m_vs_11m it is $132$).

2. The multi-task generalization setting proposed by MADT adopts a task set with similar input shapes (i.e., the observation size range is 25-48), which makes the above issue less significant.

3. Our implemented baselines (including ODIS, BC-t, and raised UPDeT variants) decompose input with particularly designed observation and state encoders to handle the varying shape issue better. It means our baseline design is generally fair and comparable. The benefits of our encoding method can be seen from the comparison between MADT and BC-t.

To validate the observed phenomena, we additionally conduct a simple experiment where the input shapes of the task set are restricted to a small range. The results are shown in Table 11. MADT generalizes better in this small task set, while ODIS can still outperform MADT.

Table 12: Average test win rates of ODIS in the marine-hard task set with expert data of 2000 trajectories (ODIS-2000) and 1000 trajectories (ODIS-1000).

|  | Task | ODIS-2000 | ODIS-1000 |
|---|---|---|---|
| Source tasks | 3m | $98.4 \pm 2.7$ | $\mathbf{100.0 \pm 0.0}$ |
|  | 5m_vs_6m | $53.9 \pm 5.1$ | $\mathbf{60.0 \pm 3.6}$ |
|  | 9m_vs_10m | $\mathbf{80.4 \pm 8.7}$ | $80.0 \pm 21.8$ |
| Unseen tasks | 4m | $\mathbf{95.3 \pm 3.5}$ | $92.5 \pm 4.7$ |
|  | 5m | $89.1 \pm 10.0$ | $\mathbf{90.0 \pm 11.1}$ |
|  | 10m | $\mathbf{93.8 \pm 2.2}$ | $74.4 \pm 27.8$ |
|  | 12m | $\mathbf{58.6 \pm 11.8}$ | $56.9 \pm 25.1$ |
|  | 7m_vs_8m | $25.0 \pm 15.1$ | $\mathbf{33.8 \pm 17.7}$ |
|  | 8m_vs_9m | $19.6 \pm 6.0$ | $\mathbf{32.5 \pm 10.8}$ |
|  | 10m_vs_11m | $\mathbf{42.2 \pm 7.2}$ | $36.2 \pm 25.8$ |
|  | 10m_vs_12m | $1.6 \pm 1.6$ | $\mathbf{3.8 \pm 5.0}$ |
|  | 13m_vs_15m | $\mathbf{2.3 \pm 2.6}$ | $0.0 \pm 0.0$ |

## H PERFORMANCE ON DIFFERENT SIZES OF DATASETS

When constructing the offline datasets, we choose the number of trajectories mainly based on recent offline MARL works (Yang et al., 2021) and the single-agent offline RL benchmark D4RL (Fu et al., 2020). As datasets in D4RL are transition-based, we estimate the corresponding trajectory numbers based on the average episode length in our evaluated benchmarks. We finally chose a trajectory number of 2000 for each task to compose multi-task offline datasets. We conduct additional experiments in the expert dataset of the marine-hard task set to evaluate ODIS with datasets of 1000 trajectory numbers. The results are exhibited in Table 12. We find that ODIS-1000 can also present a good performance, indicating that ODIS is not very sensitive to the data size.

Table 13: Test win rates of online QMIX algorithms in all test tasks. We abbreviate asymmetric tasknames for simplicity. For example, the task name "5m6m" denotes the SMAC map "5m_vs_6m".

| Task | 3m | 4m | 5m | 6m | 7m | 8m | 9m |
|---|---|---|---|---|---|---|---|
| QMIX win rate % | 99.1 | 98.2 | 99.4 | 99.9 | 99.3 | 99.2 | 99.4 |

| Task | 10m | 11m | 12m | 5m6m | 6m7m | 7m8m | 8m9m |
|---|---|---|---|---|---|---|---|
| QMIX win rate % | 99.2 | 99.4 | 99.6 | 71.9 | 74.3 | 88.8 | 93.1 |

| Task | 9m10m | 10m11m | 10m12m | 13m15m | 1s3z | 1s4z | 1s5z |
|---|---|---|---|---|---|---|---|
| QMIX win rate % | 94.3 | 97.3 | 42.7 | 64.6 | 98.5 | 97.5 | 95.3 |

| Task | 2s3z | 2s4z | 2s5z | 3s3z | 3s4z | 3s5z | 4s3z |
|---|---|---|---|---|---|---|---|
| QMIX win rate % | 96.0 | 95.1 | 94.4 | 95.1 | 87.0 | 95.1 | 94.6 |

| Task | 4s4z | 4s5z | | | | | |
|---|---|---|---|---|---|---|---|
| QMIX win rate % | 84.1 | 91.6 | | | | | |

# I    ADDITIONAL RESULTS ON MULTI-TASK OFFLINE LEARNING IN SMAC

As stated in the experiments section, we conduct experiments in three designed task sets called marine-easy, marine-hard, and stalker-zealot, respectively. As we have presented the empirical results in the marine-hard task set in Table 1, we here exhibit the results in the marine-easy and stalker-zealot task sets. The empirical results in the marine-easy task set are shown in Table 14, and the empirical results in the stalker-zealot task set are shown in Table 15. As a reference, we also exhibit the online QMIX performances in all tasks in Table 13. We find that ODIS generally outperforms other baselines in most source and unseen tasks. A specific case is that the performance of ODIS is not so promising in medium-replay data because the medium-replay data of marine-easy and stalker-zealot has very low qualities and thus hinders ODIS from discovering effective coordination skills. Despite all that, ODIS can still reach a similar performance compared to other baselines, indicating the validity of the ODIS algorithm. In addition, ODIS can achieve comparable performances to online QMIX in some unseen tasks with zero-shot generalization, showing the effectiveness of our proposed method.

Table 14: Average test win rates in the marine-easy task set with different data qualities. Results of BC-best stand for the best test win rates between BC-t and BC-r (the same to Table 15).

| Task | Expert | | | | Medium | | | |
|---|---|---|---|---|---|---|---|---|
| | BC-best | UPDeT-l | UPDeT-m | ODIS | BC-best | UPDeT-l | UPDeT-m | ODIS |
| Source tasks | | | | | | | | |
| 3m | 94.5 ± 4.6 | 77.4 ± 16.8 | 83.6 ± 12.6 | **97.7 ± 2.6** | **67.2 ± 4.7** | 40.7 ± 19.7 | 60.2 ± 29.9 | 57.8 ± 9.2 |
| 5m | 94.4 ± 7.6 | 34.8 ± 31.2 | 74.8 ± 22.9 | **95.3 ± 5.2** | 79.2 ± 5.9 | 70.2 ± 9.3 | 67.8 ± 5.9 | **82.8 ± 5.2** |
| 10m | 86.1 ± 22.7 | 48.2 ± 39.6 | 83.6 ± 19.2 | **88.3 ± 20.3** | 63.1 ± 7.2 | 35.9 ± 10.5 | 48.8 ± 7.9 | **71.9 ± 6.6** |
| Unseen Tasks | | | | | | | | |
| 4m | **91.2 ± 1.6** | 18.8 ± 4.1 | 53.0 ± 32.3 | 90.6 ± 7.0 | 62.5 ± 11.6 | 26.6 ± 23.4 | 41.7 ± 17.4 | **63.3 ± 16.1** |
| 6m | 75.3 ± 22.6 | 9.1 ± 6.7 | 37.9 ± 8.6 | **79.7 ± 17.5** | 86.0 ± 4.7 | 40.2 ± 13.2 | 75.8 ± 22.7 | **89.8 ± 17.6** |
| 7m | 70.3 ± 11.0 | 28.9 ± 6.4 | 44.2 ± 13.2 | **72.7 ± 16.9** | **100.0 ± 0.0** | 39.8 ± 19.4 | 65.2 ± 25.2 | 96.1 ± 1.4 |
| 8m | 74.7 ± 16.5 | 31.0 ± 11.4 | 51.7 ± 26.2 | **80.9 ± 14.4** | 96.9 ± 2.2 | 18.2 ± 7.8 | 88.4 ± 13.7 | **97.7 ± 2.6** |
| 9m | 97.7 ± 2.6 | 28.1 ± 15.5 | 76.3 ± 13.4 | **99.2 ± 1.4** | 78.9 ± 11.8 | 35.9 ± 14.7 | 64.8 ± 35.6 | **87.5 ± 2.2** |
| 11m | 83.3 ± 11.8 | 20.3 ± 4.7 | 53.6 ± 22.4 | **83.6 ± 12.4** | 42.2 ± 4.7 | 26.6 ± 7.8 | 23.4 ± 11.8 | **64.7 ± 3.1** |
| 12m | 56.7 ± 30.0 | 16.3 ± 6.4 | 44.3 ± 22.8 | **70.3 ± 30.2** | 29.7 ± 23.4 | 5.5 ± 7.8 | 13.5 ± 11.7 | **41.4 ± 6.0** |
| | Medium-Expert | | | | Medium-Replay | | | |
| Seen Tasks | | | | | | | | |
| 3m | 81.3 ± 18.8 | 43.7 ± 30.9 | 48.4 ± 36.8 | **89.8 ± 9.7** | 77.8 ± 3.2 | 44.2 ± 13.5 | 29.7 ± 10.0 | **79.7 ± 4.7** |
| 5m | 74.0 ± 2.9 | 45.6 ± 27.1 | 64.1 ± 17.9 | **83.7 ± 16.0** | 5.5 ± 5.6 | **16.1 ± 27.9** | 6.2 ± 10.8 | 3.1 ± 5.4 |
| 10m | 90.6 ± 3.1 | 42.3 ± 26.0 | 68.8 ± 23.8 | **93.8 ± 4.4** | 0.0 ± 0.0 | 0.0 ± 0.0 | 0.0 ± 0.0 | 0.0 ± 0.0 |
| Unseen Tasks | | | | | | | | |
| 4m | 35.2 ± 38.0 | 53.5 ± 31.2 | 43.7 ± 25.0 | **57.8 ± 18.8** | **67.2 ± 4.7** | 25.5 ± 18.2 | 25.0 ± 22.6 | 25.0 ± 5.4 |
| 6m | 42.2 ± 1.6 | 57.0 ± 25.1 | 47.7 ± 30.0 | **76.0 ± 6.0** | 7.8 ± 10.2 | **12.1 ± 21.0** | 0.0 ± 0.0 | 3.1 ± 5.4 |
| 7m | 65.6 ± 16.4 | 32.0 ± 30.3 | 57.8 ± 32.9 | **66.4 ± 14.6** | **0.8 ± 1.4** | 0.0 ± 0.0 | 0.0 ± 0.0 | 0.0 ± 0.0 |
| 8m | 40.3 ± 42.6 | 35.9 ± 19.6 | 40.6 ± 19.3 | **43.8 ± 11.5** | 0.8 ± 1.4 | 0.8 ± 1.4 | 0.0 ± 0.0 | **1.6 ± 1.6** |
| 9m | 70.8 ± 16.6 | **75.0 ± 9.4** | 47.7 ± 24.8 | 73.4 ± 16.2 | **0.8 ± 1.4** | 0.0 ± 0.0 | 0.0 ± 0.0 | 0.0 ± 0.0 |
| 11m | 55.5 ± 12.4 | 51.6 ± 35.9 | **85.9 ± 14.2** | 68.8 ± 20.3 | 0.0 ± 0.0 | 0.0 ± 0.0 | 0.0 ± 0.0 | 0.0 ± 0.0 |
| 12m | 29.7 ± 29.8 | 19.7 ± 20.0 | 46.1 ± 15.5 | **62.5 ± 8.0** | 0.0 ± 0.0 | 0.0 ± 0.0 | 0.0 ± 0.0 | 0.0 ± 0.0 |

Table 15: Average test win rates in the stalker-zealot task set with different data qualities.

| Task | Expert | | | | Medium | | | |
|---|---|---|---|---|---|---|---|---|
| | BC-best | UPDeT-l | UPDeT-m | ODIS | BC-best | UPDeT-l | UPDeT-m | ODIS |
| Source tasks | | | | | | | | |
| 2s3z | 93.1 ± 4.6 | 53.1 ± 39.1 | 50.0 ± 33.4 | **97.7 ± 2.6** | 48.8 ± 9.8 | 30.6 ± 12.7 | 35.0 ± 23.0 | **49.2 ± 8.4** |
| 2s4z | **78.1 ± 8.1** | 48.4 ± 24.3 | 23.4 ± 26.6 | 60.9 ± 6.8 | 12.5 ± 8.1 | 28.8 ± 4.1 | 18.8 ± 10.3 | **32.8 ± 12.2** |
| 3s5z | **92.5 ± 4.2** | 40.6 ± 11.5 | 17.2 ± 19.8 | 87.5 ± 9.6 | 24.4 ± 12.4 | 15.0 ± 10.3 | 25.6 ± 24.2 | **28.9 ± 6.8** |
| Unseen Tasks | | | | | | | | |
| 1s3z | 45.6 ± 23.8 | 26.6 ± 25.0 | 1.6 ± 1.6 | **76.6 ± 3.5** | 21.9 ± 37.6 | 33.1 ± 18.0 | 3.8 ± 5.0 | **41.4 ± 18.8** |
| 1s4z | **60.0 ± 32.3** | 37.5 ± 31.9 | 26.6 ± 19.3 | 17.2 ± 10.5 | 6.2 ± 7.7 | 35.0 ± 7.2 | 2.5 ± 3.6 | **50.7 ± 7.5** |
| 1s5z | **45.6 ± 26.9** | 14.8 ± 13.9 | 29.7 ± 26.4 | 2.5 ± 2.3 | 3.1 ± 2.6 | 13.1 ± 11.4 | 5.0 ± 4.2 | **14.1 ± 8.4** |
| 2s5z | **75.6 ± 11.9** | 27.3 ± 19.3 | 23.4 ± 22.2 | 27.3 ± 6.0 | 14.4 ± 9.0 | 17.5 ± 9.2 | 16.9 ± 14.1 | **32.0 ± 4.6** |
| 3s3z | 80.6 ± 9.1 | 49.2 ± 25.8 | 20.3 ± 10.9 | **89.1 ± 5.2** | **45.6 ± 14.6** | 23.8 ± 6.7 | 24.4 ± 28.6 | 23.4 ± 9.2 |
| 3s4z | 92.5 ± 5.1 | 59.4 ± 16.4 | 12.5 ± 19.9 | **96.9 ± 2.2** | 40.0 ± 19.0 | 17.5 ± 10.0 | 28.8 ± 31.6 | **50.8 ± 15.5** |
| 4s3z | **67.5 ± 19.8** | 50.8 ± 24.8 | 6.2 ± 4.4 | 64.1 ± 13.0 | **28.8 ± 26.4** | 3.1 ± 4.0 | 11.2 ± 18.0 | 13.3 ± 7.5 |
| 4s4z | 53.1 ± 18.4 | 41.4 ± 16.0 | 7.8 ± 13.5 | **79.7 ± 10.9** | **20.0 ± 12.0** | 1.9 ± 2.5 | 1.2 ± 1.5 | 12.5 ± 7.0 |
| 4s5z | 40.6 ± 19.1 | 28.1 ± 17.0 | 5.5 ± 7.8 | **86.7 ± 12.6** | **14.4 ± 8.5** | 5.0 ± 5.4 | 5.6 ± 8.5 | 7.0 ± 4.1 |
| 4s6z | 48.1 ± 23.8 | 10.9 ± 7.2 | 4.7 ± 6.4 | **88.3 ± 8.4** | **3.8 ± 3.6** | 2.5 ± 2.3 | 1.9 ± 2.5 | 1.6 ± 1.6 |
| | Medium-Expert | | | | Medium-Replay | | | |
| Seen Tasks | | | | | | | | |
| 2s3z | 57.5 ± 25.1 | **59.4 ± 20.8** | 57.5 ± 27.1 | 58.6 ± 15.5 | 3.1 ± 2.6 | 6.9 ± 10.7 | 14.4 ± 13.2 | **15.6 ± 18.2** |
| 2s4z | 37.5 ± 15.3 | 32.0 ± 11.1 | **53.1 ± 24.6** | 41.4 ± 7.8 | 5.2 ± 7.4 | 6.2 ± 9.5 | **12.5 ± 9.7** | 7.8 ± 5.2 |
| 3s5z | **63.1 ± 13.3** | 18.0 ± 11.8 | 35.0 ± 23.5 | 41.4 ± 18.5 | 31.3 ± 6.3 | 5.0 ± 7.0 | **20.0 ± 16.6** | 18.8 ± 3.1 |
| Unseen Tasks | | | | | | | | |
| 1s3z | 55.6 ± 37.7 | 23.4 ± 14.1 | 4.4 ± 8.8 | **72.7 ± 12.2** | **24.0 ± 15.4** | 8.1 ± 10.4 | 0.0 ± 0.0 | 21.1 ± 20.4 |
| 1s4z | 25.0 ± 30.7 | 35.2 ± 18.0 | 11.9 ± 9.8 | **44.5 ± 20.3** | 2.1 ± 2.9 | 6.9 ± 13.8 | **7.5 ± 10.0** | 6.2 ± 7.7 |
| 1s5z | 14.4 ± 19.4 | 16.4 ± 11.6 | 3.8 ± 4.6 | **42.2 ± 31.4** | 7.3 ± 6.4 | 1.9 ± 3.8 | **11.9 ± 9.6** | 7.8 ± 6.4 |
| 2s5z | 26.9 ± 20.2 | 5.5 ± 7.8 | 37.5 ± 22.5 | **43.0 ± 10.7** | 12.5 ± 15.5 | 4.4 ± 8.8 | **20.0 ± 16.8** | 14.1 ± 8.1 |
| 3s3z | 35.6 ± 18.0 | 7.0 ± 3.4 | 33.8 ± 15.0 | **50.0 ± 13.3** | **35.4 ± 12.1** | 5.6 ± 7.0 | 17.5 ± 12.3 | 25.0 ± 20.1 |
| 3s4z | **74.4 ± 16.3** | 3.1 ± 3.1 | 43.1 ± 20.7 | 52.3 ± 9.5 | **20.8 ± 9.0** | 5.0 ± 8.5 | 15.6 ± 11.2 | 19.5 ± 16.6 |
| 4s3z | **69.8 ± 7.8** | 0.0 ± 0.0 | 23.8 ± 21.0 | 17.2 ± 7.2 | **17.7 ± 5.3** | 11.2 ± 13.9 | 11.2 ± 15.0 | 8.6 ± 14.9 |
| 4s4z | **41.9 ± 14.9** | 3.1 ± 3.8 | 10.6 ± 13.8 | 20.3 ± 6.8 | **15.6 ± 6.8** | 3.8 ± 3.6 | 5.6 ± 9.8 | 4.7 ± 8.1 |
| 4s5z | 17.3 ± 5.3 | 0.8 ± 1.4 | 11.9 ± 16.1 | **21.9 ± 2.2** | 1.0 ± 1.5 | 3.1 ± 4.0 | **10.6 ± 19.7** | 0.8 ± 1.4 |
| 4s6z | 13.8 ± 3.2 | 0.8 ± 1.4 | 5.0 ± 8.5 | **18.0 ± 5.1** | 0.0 ± 0.0 | 2.5 ± 5.0 | **6.9 ± 13.8** | 2.3 ± 4.1 |

