# OpenReview forum: "Discovering Generalizable Multi-agent Coordination Skills from Multi-task Offline Data"
_ICLR.cc/2023/Conference — ICLR 2023 poster_

### Official Review · Reviewer_1rgQ · 2022-10-13

**Confidence:** 4
**Correctness:** 4
**Technical Novelty And Significance:** 3
**Empirical Novelty And Significance:** 3
**Recommendation:** 8

**Clarity, Quality, Novelty And Reproducibility:**

The paper is clearly written, although the appendix is doing a lot of heavy lifting for architecture design, training details, extra experiments, dataset details, etc. The main body of the paper is self-contained for describing the methods and the experiments, although the appendix is frequently referenced and used.

The work appears to be original/novel and high quality. Comparisons to recent work are highlighted by the authors, and differences are clearly discussed and experiments are conducted to highlight performance gaps.

**Strength And Weaknesses:**

Strengths:
* ODIS is shown to perform better than baselines in multi-agent problems.
* ODIS generalizes to new state-action space configurations, such as adding more Marines in the StarCraft II micro-control domains.
* The appendix has a wealth of detail on the experimental design and results, additional results and experiments, and network architecture and training details.
* The approach is relatively clear and the loss functions are well-introduced and explained.
* Contributions are evaluated with ablation studies across new objectives, showing the importance of the consistent loss that is introduced as part of the method.

Weaknesses:
* The experimental setup and results are jumbled together in a somewhat confusing way. Section 5.1 introduces the baselines and describes how they relate to ODIS, and at the same time it is delivering results and discussion. There is a lot of information for the reader to make sense of all at once, without having a full-picture of the results or baselines.
* The experiments and introduction mention an additional navigation task, but it is not present in the main body of the paper.
* The main body of the paper references and discusses a result that is only present in the appendix (Figure 7).

**Summary Of The Paper:**

The paper presents ODIS, a new approach to learning policies for multi-agent, multi-task RL problems via skill-discovery and conditioning. Leveraging transformer models, the proposed approach can learn state-action-space invariant policies from offline data, and through additional loss functions the policies are encouraged to learn distinctive "coordination skills" that can be used to aid in action generation. Experimental evaluations show ODIS often outperforms simple BC baselines as well as other MARL approaches using comparable optimization steps.

**Summary Of The Review:**

The paper presents an interesting approach that is clearly described and shows strong empirical performance relative to baselines that are given in the paper. The main weakness of the paper is that many of the relevant details of the method (network architecture, dataset/training details, experimental design/results, comparison to contemporary work, etc.) are relegated to an appendix. However, the main body of the paper does not often suffer for moving so much information to the appendix, and overall the submission is strong.

---

> ### Author Response · Authors · 2022-11-13
> **Response to Reviewer 1rgQ**
>
> Thank you for your inspiring and thorough comments! We are encouraged by your careful review and constructive suggestions. Here are our responses to your proposed weaknesses in our papers.
>
> ---
>
> **Weakness 1**: The experimental setup and results are jumbled together in a somewhat confusing way in Section 5.1. There is a lot of information for the reader to make sense of all at once without having a full picture of the results or baselines.
>
> **Response 1**: We are dedicated to improving the readability according to your suggestions. We notice that it is not easy to clearly describe the environment information, dataset properties, and all baselines in limited spaces. To polish this subsection, we reorder the content to divide it into three parts, including introducing all baselines, the experimental setup and results for SMAC, and a brief introduction to cooperative navigation experiments, respectively.
>
> ---
>
> **Weakness 2**: The experiments and introduction mention an additional navigation task, but it is not present in the main body of the paper.
>
> **Response 2**: We mainly conduct our experiments on SMAC tasks, as it contains diverse and complex tasks for sufficient evaluations across generalization, semantic visualization, etc. To better validate the effectiveness, we additionally evaluate our method in cooperative navigation tasks. Due to the page limit, we have to defer these relative experiments to our appendix. This part is relatively independent, and we expect that readers will not miss the major results and conclusions without reading this part. To make the main body of our paper more self-contained, we remove the reference to cooperative navigation tasks in other sections except for the last paragraph of Section 5.1.
>
> ---
>
> **Weakness 3**: The main body of the paper references and discusses a result that is only present in the appendix (Figure 7).
>
> **Response 3**: We use Figure 7 to illustrate that the performance of ODIS is comparable to single-task offline MARL methods. Although the evaluation of single-task performance is not the primary part of our paper, we think that the experiments can help express the effectiveness of ODIS. To make the experiments section more coherent, we remove the direct reference to Figure 7 and make some revisions, where we only shortly discuss the experiments and provide the reference to the related appendix.
>
> ---
>
> We are encouraged by your positive comments on our paper. We believe that your reviews can help us improve the readability and make the work clearer to general readers. We are happy to hear further suggestions from you.

---

### Official Review · Reviewer_PuN1 · 2022-10-24

**Confidence:** 2
**Correctness:** 3
**Technical Novelty And Significance:** 2
**Empirical Novelty And Significance:** 2
**Recommendation:** 5

**Clarity, Quality, Novelty And Reproducibility:**


<Methodology>

1. The process of skill discovery seems to be nothing but imitation learning with latent factor modeling. Although current studies claim z to be one of the possible skills, there is no evidence that z_i implies a specific thawing pattern.

2. The skill discovery module and Q-mix style MARL algorithms seem to be trained separately. What is the reason for this, and is there any issue arising from this separate training?

3. It is ambiguous where $\hat{q}_i$ is used. It seems that $\hat{q}_i$ amortizes the global skill discovery encoder. The amortized encoder may have lower generalization capability, especially when the distribution shift occurs during multi-task transfer.


<Experiments>

When the trained model from the source data is tested with unseen tasks, the unseen task of a similar scale, the performance drops sharply. For example, when a model trained at 9m10m is tested at 10m11m, the performance drops by more than 50%. It may be hard to say that the proposed model can effectively transfer to unseen tasks.


**Strength And Weaknesses:**

To conduct multi-task learning, the current study utilizes a kind of high-level action called a skill that co-exists in multiple tasks. In addition, to conduct offline learning, the current study uses imitation learning with a conservative Q-value estimation technique. Each technique is already an existing methodology, and it seems this study used these two techniques simultaneously to achieve two purposes.

However, this study did not formalize or discuss additional issues that may arise when both cases occur simultaneously. Both problems have a common issue of distribution shift, but no means to overcome this problem is specifically suggested.

**Summary Of The Paper:**

Although there have been studies that extended MARL into offline learning concepts and multi-tasking learning, no studies have attempted both simultaneously. Thus, the current research is meaningful in that it tries to tackle both offline and multi-task learning simultaneously in MARL. The principal methodology that the current study uses is inferring the latent vector that can possibly represent the essential skills for inducing coordination among agents and using these inferred skills in independent decision-making.

**Summary Of The Review:**

Although it is a very challenging study that combines offline learning and multi-task learning with MARL, the methodological novelty is not great. In addition, too much overstatement of the latent variable z.

---

> ### Author Response · Authors · 2022-11-13
> **Response to Reviewer PuN1 (Part 2/2)**
>
> **Q5**: What is the reason for separate training? Is there any issue arising from this separate training?
>
> **A5**: Separate training faces the problem of no joint optimization. However, it is not an issue in our methodology since the low-level policy training (i.e., the skill discovery) does not rely on the high-level policy. Performing joint training will make the optimization problem more complex without significant advantages.
>
> ---
>
> **Q6**: It is ambiguous where $\hat{q}_i$ is used. The amortized encoder $\hat{q}_i$ may have lower generalization capability.
>
> **A6**: The observation encoder $\hat{q}_i$ is a component of the individual coordination policy, which calculates a representation from local information. $\hat{q}_i$ may not generalize as well as the global state encoder $q$. However, we **cannot access global information during decentralized execution**. Thus we introduce the consistent loss to maintain the consistency between these two encoders with representation learning. This approach helps improve the generalization capability as shown in our ablation studies in Figure 4(b).
>
> ---
>
> **Q7**: The performance drops in unseen tasks. It may be hard to say that the proposed model can effectively transfer to unseen tasks.
>
> **A7**: The effectiveness of transferring to unseen tasks is shown when comparing to other baselines. ODIS can directly perform well in many unseen tasks without significant performance decreases. In some difficult tasks, ODIS may not maintain high performance with the direct zero-shot transfer. ODIS outperforms these baselines in most tasks and none of these baselines can effectively transfer to difficult unseen tasks.
>
> ---
>
> We hope that our clarifications can help you better understand the contributions and methodology of our paper. Please feel free to add a new comment if there are still unclear parts. We are happy to answer any further questions and sincerely thank you for your careful reviews.
>
> [1] Wenxuan Zhou, Sujay Bajracharya, and David Held. PLAS: latent action space for offline reinforcement learning. In Conference on Robot Learning, pp. 1719–1735, 2020.
>
> [2] Anurag Ajay, Aviral Kumar, Pulkit Agrawal, Sergey Levine, and Ofir Nachum. OPAL: offline primitive discovery for accelerating offline reinforcement learning. In International Conference on Learning Representations, 2021.

---

> ### Author Response · Authors · 2022-11-13
> **Response to Reviewer PuN1 (Part 1/2)**
>
> Thank you for your thoughtful reviews and constructive suggestions! We provide some clarifications for your comments, and we hope they can help address your concerns about our paper.
>
> ---
>
> **Q1**: The skill discovery seems to be imitation learning with latent factor modeling.
>
> **A1**: We would like to emphasize the difference between our method and imitation learning. Imitation learning, even with latent factor modeling, is to recover the behavior policy, but does not mean to beyond. With the imitation purpose, the latent factor modeling aims at further maximizing the likelihood of the imitation. However, our state encoder provides a set of coordination skills, from which the skill can be chosen by the high-level policy to achieve a performance that is beyond the behavior policy. Our method contains parts of objectives/tricks as previous ones, which serve a very different purpose and, therefore, should not weaken our novelty.
>
> ---
>
> **Q2**: Although current studies claim $z$ to be one of the possible skills, there is no evidence that $z_i$ implies a specific thawing pattern. There is too much overstatement of the latent variable $z$.
>
> **A2**: Our coordination skill is learned to derive particular behaviors under distinguished tasks and conditions. To realize this, we adopt the VAE-style objective for skill discovery that is widely used in offline latent action learning methods [1, 2]. We empirically find that the discovered skill can imply specific patterns that differ from the training multi-task data. In Figure 3 of our paper, we anticipate the semantics of different skills in test episodes and find that the chosen actions with given skills highly corresponded to our anticipations. Our experiments show that the discovered skills automatically exhibit different semantics, although we cannot specify their patterns. We have modified Figure 3 and related discussions to avoid overstating the discovered skills as our anticipations.
>
> ---
>
> **Q3**: The current study seems to use imitation learning with a conservative Q-value estimation technique simultaneously to achieve two purposes. Each technique is already an existing methodology.
>
> **A3**: Our study focuses on the main purpose of extracting and reusing coordination skills shared in multi-task data to improve generalization but not two separate purposes. The challenge of generalizable offline MARL cannot be solved by either imitation learning or conservative Q-value estimation. Even though we consider combining these two techniques in offline MARL, it is non-trivial to adopt them simultaneously to learn a generalizable policy. Moreover, our developed hierarchical policy structure and training algorithm is novel and distinguished from previous works.
>
> ---
>
> **Q4**: This study did not formalize or discuss additional issues that may arise when both cases occur simultaneously. Both problems have a common issue of distribution shift, but no means to overcome this problem is specifically suggested.
>
> **A4**: The co-existing of unsupervised skill discovery and conservative Q-learning can help alleviate the distribution shift issue jointly. As imitation learning only executes the behavior policy, introducing a high-level policy on the coordination skill can help select other better actions when facing distribution shift. On the other hand, offline RL training faces the problem of extrapolation errors when updating Q-values with out-of-distribution data. We reduce the high-level policy space to a unified coordination skill space and ensure the coordination skill can correspond to actions in offline data, which also helps tackle the distribution shift.

---

### Official Review · Reviewer_e7iL · 2022-10-25

**Confidence:** 3
**Correctness:** 3
**Technical Novelty And Significance:** 3
**Empirical Novelty And Significance:** 3
**Recommendation:** 6

**Clarity, Quality, Novelty And Reproducibility:**

The paper is well-written, with clear motivation for the problem and the method, and is also easy to follow. The authors conduct extensive experiments on several maps from SMAC in comparison to recent baselines to validate the effectiveness of ODIS. The setting is new and the proposed method is also reasonable.

**Strength And Weaknesses:**

### Strengths
The paper tackles an important problem in MARL - generalization to tasks with varying agents and targets based on the recent success of offline MARL algorithms that leverage the offline dataset (compared to a fully online way). The setting is novel and new. The proposed method is also reasonable and clearly written, which is motivated well. Figure 1 does a good job in motivating the problem and the proposed method, where ODIS is responsible for generalizing to the unseen 10m task given offline data from source tasks like 5m and 8m via discovering coordination skills and learning a coordination policy. The paper is clearly written and easy to follow.

### Weaknesses
One minor concern for the paper is to evaluate ODIS in maps that is more difficult (like corridor and 2c_vs_64zg), as the experiments are mainly focused on easy maps.

**Summary Of The Paper:**

The paper aims to discover generalizable multi-agent coordination skills from multi-task offline data (multi-task marl with offline data). The insight is that we can extract universal skills for coordination from offline multi-agent multi-task data. The authors propose the ODIS algorithm which consists of two stages. Firstly, task-invariant coordination skills are extracted from the offline multi-task data, and it learns to delineate different agent behaviors with the discovered coordination skills. Secondly, a coordination policy is trained to choose optimal c coordination skills under CTDE. The authors conduct extensive experiments based on several maps from SMAC to validate the effectiveness of ODIS.

**Summary Of The Review:**

In all, the paper studies an intersting and new setting, which aims to improve generalization in MARL via offline multi-task multi-agent data. The proposed method is also reasonable, with significant improvements over baselines.

---

> ### Author Response · Authors · 2022-11-13
> **Response to Reviewer e7iL**
>
> Thank you for your careful and constructive comments! We have prepared the following experimental results and comments for your proposed weakness, and we hope they can relieve your concern.
>
> ---
>
> **Weakness**: Evaluate ODIS in more difficult maps (like corridor and 2c_vs_64zg), as the experiments are mainly focused on easy maps.
>
> **Response**:
>
> Actually, we have already evaluated ODIS in a few maps that are not easy such as several asymmetric maps in the marine-hard task set. We consider conducting experiments in your mentioned maps like corridor and 2c_vs_64zg. We find that corridor is an extremely hard map for an online QMIX algorithm to acquire enough performance, which **hinders data collection for offline RL experiments**. Therefore, we mainly create a task set of maps similar to 2c_vs_64zg and call it the colossus task set. It contains five tasks, where two are source tasks and others are unseen tasks. The data collection scheme is the same as previous experiments in our paper. We present the results in the following table.
>
> | Task                               | ODIS (ours)     | BC-best         | UPDeT-m   | UPDeT-l   |
> | ---------------------------------- | --------------- | --------------- | --------- | --------- |
> | 2c_vs_64zg-expert (source)         | **65.6 ± 3.1**  | 64.1 ± 10.9     | 0.0 ± 0.0 | 3.1 ± 5.4 |
> | 3c_vs_100zg-expert (source)        | **67.2 ± 10.9** | 48.4 ± 32.8     | 0.8 ± 1.4 | 3.1 ± 3.1 |
> | 2c_vs_72zg-expert (unseen)         | **31.2 ± 3.1**  | 12.5 ± 3.1      | 0.0 ± 0.0 | 3.9 ± 6.8 |
> | 3c_vs_96zg-expert (unseen)         | **93.8 ± 0.0**  | 39.1 ± 7.2      | 0.0 ± 0.0 | 4.7 ± 4.7 |
> | 3c_vs_108zg-expert (unseen)        | **76.6 ± 1.6**  | 26.6 ± 7.8      | 0.0 ± 0.0 | 0.8 ± 1.4 |
> |                                    |                 |                 |           |           |
> | 2c_vs_64zg-medium (source)         | **15.6 ± 2.9**  | 14.1 ± 3.1      | 1.0 ± 1.5 | 0.0 ± 0.0 |
> | 3c_vs_100zg-medium (source)        | **31.2 ± 6.1**  | 15.6 ± 15.6     | 0.0 ± 0.0 | 0.0 ± 0.0 |
> | 2c_vs_72zg-medium (unseen)         | **5.2 ± 3.9**   | 1.6 ± 1.6       | 0.0 ± 0.0 | 0.0 ± 0.0 |
> | 3c_vs_96zg-medium (unseen)         | 31.2 ± 6.4      | **35.9 ± 29.7** | 0.0 ± 0.0 | 0.0 ± 0.0 |
> | 3c_vs_108zg-medium (unseen)        | **16.7 ± 5.3**  | 3.1 ± 1.5       | 0.0 ± 0.0 | 0.0 ± 0.0 |
> |                                    |                 |                 |           |           |
> | 2c_vs_64zg-medium-expert (source)  | **21.9 ± 9.4**  | 0.0 ± 0.0       | 0.0 ± 0.0 | 0.0 ± 0.0 |
> | 3c_vs_100zg-medium-expert (source) | **12.5 ± 6.2**  | 4.7 ± 4.7       | 0.0 ± 0.0 | 0.0 ± 0.0 |
> | 2c_vs_72zg-medium-expert (unseen)  | **3.1 ± 0.0**   | 0.0 ± 0.0       | 0.0 ± 0.0 | 0.0 ± 0.0 |
> | 3c_vs_96zg-medium-expert (unseen)  | **40.6 ± 18.8** | 20.3 ± 4.7      | 0.0 ± 0.0 | 0.0 ± 0.0 |
> | 3c_vs_108zg-medium-expert (unseen) | 6.2 ± 3.1       | **9.4 ± 6.2**   | 0.0 ± 0.0 | 0.0 ± 0.0 |
> |                                    |                 |                 |           |           |
> | 2c_vs_64zg-medium-replay (source)  | **3.1 ± 1.5**   | **3.1 ± 3.1**   | 0.0 ± 0.0 | 0.0 ± 0.0 |
> | 3c_vs_100zg-medium-replay (source) | **40.6 ± 15.9** | 17.2 ± 10.9     | 0.0 ± 0.0 | 0.0 ± 0.0 |
> | 2c_vs_72zg-medium-replay (unseen)  | 0.0 ± 0.0       | 0.0 ± 0.0       | 0.0 ± 0.0 | 0.0 ± 0.0 |
> | 3c_vs_96zg-medium-replay (unseen)  | **46.9 ± 13.5** | 10.9 ± 1.6      | 0.8 ± 1.4 | 0.0 ± 0.0 |
> | 3c_vs_108zg-medium-replay (unseen) | **43.8 ± 19.3** | 14.1 ± 14.1     | 0.0 ± 0.0 | 0.0 ± 0.0 |
>
> We find that ODIS can outperform other baselines in most tasks with different data qualities. Our transformer-based behavior cloning methods can also perform well in some tasks, while multi-task MARL baselines fail to learn a good policy in this task set. The results indicate that discovering effective coordination skills for high-level coordination policy learning can be helpful for multi-task generalization.
>
> ---
>
> We hope that our additional experiments can address your concerns about our paper. Please feel free to add a comment if you have further questions.

---

### Official Review · Reviewer_E6Nd · 2022-10-28

**Confidence:** 3
**Correctness:** 3
**Technical Novelty And Significance:** 3
**Empirical Novelty And Significance:** 4
**Recommendation:** 8

**Clarity, Quality, Novelty And Reproducibility:**

This paper presents a novel method in an important and relatively unexplored area. It is clearly written and well organized, and has the experiments and ablations necessary for a high quality submission



**Strength And Weaknesses:**

Strengths:

1) This paper addresses the intersection of three important areas of research - multi-agent, multi-task, and offline RL. It is the first paper to directly address these areas and notably outperforms single-task offline MARL methods.

2) The paper is well written, with thorough experiments, appropriate baselines, and a variety of tasks across two distinct environments.

Weaknesses:

1) The method requires access to global state to learn the skill extractor. It's not clear to me why learning observation encoders and a mixing network, as is done in the coordination policy, is not sufficient, and some intuition would be helpful.

2) The authors state that the distribution of skills is prevented from collapsing by maximizing KLD with a uniform skill distribution. While Figure 3 seems to support this, learning diverse skills automatically has historically been a difficult problem [1, 2], and I would appreciate some further discussion or ablations around this.

3) In most tasks, the results are only slightly better than behavioral cloning, even on the lower quality datasets.

4) The transformer-based architecture is closely related to [3], and I would recommend it be added to the literature review.

Questions:

1) How many evaluation trajectories are run for each result?

2) Is any parameter sharing used in the observation encoder or action decoder?

3) Are there any results comparing IQL or QPLEX on other datasets besides medium?

4) Does freezing the last layer of the skill encoder affect the performance of the skill classifier?

5) Is there a reason to not standardize the size of the medium-replay dataset, using the same construction described? This would make the results more comparable to the other datasets.

[1] Eysenbach, B., Gupta, A., Ibarz, J., & Levine, S. (2018, September). Diversity is All You Need: Learning Skills without a Reward Function. In International Conference on Learning Representations.

[2] Haarnoja, T., Hartikainen, K., Abbeel, P., & Levine, S. (2018, July). Latent space policies for hierarchical reinforcement learning. In International Conference on Machine Learning (pp. 1851-1860). PMLR.

[3] Iqbal, S., & Sha, F. (2019, May). Actor-attention-critic for multi-agent reinforcement learning. In International conference on machine learning (pp. 2961-2970). PMLR.

**Summary Of The Paper:**

This paper proposes a novel algorithm, ODIS, for learning multiple multi-agent tasks from an unlabeled offline dataset. The algorithm is broken up into two phases. In the first phase, a skill extractor is learned from global states, and the encoded skills are fed to per-agent action decoders. In the second stage, per-agent observation encoders

**Summary Of The Review:**

Overall, this paper proposes a novel idea with promising results in an important field, and it is well written and well supported. While I have some minor questions, I would recommend this paper for acceptance.

---

> ### Author Response · Authors · 2022-11-13
> **Response to Reviewer E6Nd (Part 2/2)**
>
> **Question 1**: How many evaluation trajectories are run for each result?
>
> **Answer 1**: We evaluate 32 episodes and compute the test win rates for each run. The evaluation manner is the same as the [PyMARL](https://github.com/oxwhirl/pymarl) framework, which is widely used in recent works like QMIX, QTRAN, COMA, etc.
>
> ---
>
> **Question 2**: Is any parameter sharing used in the observation encoder or action decoder?
>
> **Answer 2**: Yes. All agent-wise components, including the action decoder and the coordination policy, are parameter-sharing. We believe that the parameter-sharing technique benefit from dealing with varying agent numbers.
>
> ---
>
> **Question 3**: Are there any results comparing ICQ or QPLEX on other datasets besides medium?
>
> **Answer 3**: We additionally evaluate ICQ and QPLEX in expert and medium-expert datasets, and the results are appended to Figure 7 in the appendix. We also present these experiments' final test win rates in the following table. Like our previous experiments, 5m and 10m are two source tasks and 8m is the unseen task of ODIS, while single-task offline MARL methods, ICQ and QPLEX, use data from each single task. We find that ODIS has comparable performance in most tasks. ICQ performs well in the expert dataset due to its proposed extremely conservative learning. QPLEX fails to learn a good policy in our datasets with large extrapolation errors. ODIS can acquire comparable asymptotic performance without using any data from 8m, while the two other methods use additional data of 8m.
>
> | Task | ODIS (ours) | ICQ | QPLEX |
> | --------------------------------------- | ------------------- | ------------------ | ------------- |
> | 5m-medium | **82.8 $\pm$ 5.2**  | 76.0$\pm$6.4       | 0.0 $\pm$ 0.0 |
> | 10m-medium | **71.9 $\pm$ 6.6**  | 56.3$\pm$ 10.6     | 0.0 $\pm$ 0.0 |
> | 8m-medium (only unseen for ODIS)        | **97.7 $\pm$ 2.6**  | 80.6 $\pm$ 3.6     | 0.0 $\pm$ 0.0 |
> | 5m-expert | **95.3 $\pm$ 5.2**  | 80.6$\pm$17.2      | 0.0 $\pm$ 0.0 |
> | 10m-expert | 88.3 $\pm$ 20.3     | **95.6$\pm$ 4.2**  | 0.0 $\pm$ 0.0 |
> | 8m-expert (only unseen for ODIS)        | 80.9 $\pm$ 14.4     | **97.5 $\pm$ 3.6** | 0.0 $\pm$ 0.0 |
> | 5m-medium-expert | **83.7 $\pm$ 16.0** | 68.1$\pm$15.5      | 0.0 $\pm$ 0.0 |
> | 10m-medium-expert | **93.8 $\pm$ 4.4**  | 72.5$\pm$ 21.1     | 0.0 $\pm$ 0.0 |
> | 8m-medium-expert (only unseen for ODIS) | **43.8 $\pm$ 11.5** | 40.6 $\pm$ 20.0    | 0.0 $\pm$ 0.0 |
>
> ---
>
> **Question 4**: Does freezing the last layer of the skill encoder affect the performance of the skill classifier?
>
> **Answer 4**: Our stated skill classifier is the last layer of the state encoder. We have finished the training of the state encoder when computing the consistent loss, and the total state encoder is frozen for providing supervised signals. We are indeed learning the observation encoder here using the last layer of the state encoder to calculate the consistent loss. We have revised our method section to make it clearer.
>
> ---
>
> **Question 5**: Is there a reason not to standardize the size of the medium-replay dataset using the same construction described?
>
> **Answer 5**: The size of the medium-replay dataset is unfixed because the medium-replay data are actually the replay buffer of QMIX, depending on the training steps of a medium QMIX policy. We follow the convention of medium-replay data from the popular D4RL [5] offline RL benchmark. We believe it will be better to keep the convention rather than standardize the data size. For example, the Hopper-v2-medium-replay dataset of D4RL contains 402,000 transitions, while the corresponding typical expert/medium dataset contains 1M transitions.
>
> ---
>
> We hope our clarifications and answers can help address your concerns. We express our gratitude for your careful review and are dedicated to improving our paper according to your constructive suggestions. Please let us know if there are further questions.
>
> [1] Benjamin Eysenbach, Abhishek Gupta, Julian Ibarz, and Sergey Levine. Diversity is all you need: Learning skills without a reward function. In International Conference on Learning Representations, 2019.
>
> [2] Tuomas Haarnoja, Kristian Hartikainen, Pieter Abbeel, and Sergey Levine. Latent space policies for hierarchical reinforcement learning. In International Conference on Machine Learning, pp. 1846–1855, 2018.
>
> [3] Lili Chen, Kevin Lu, Aravind Rajeswaran, Kimin Lee, Aditya Grover, Michael Laskin, Pieter Abbeel, Aravind Srinivas, and Igor Mordatch. Decision transformer: Reinforcement learning via sequence modeling. In Advances in Neural Information Processing Systems, pp. 15084–15097, 2021.
>
> [4] Shariq Iqbal and Fei Sha. Actor-attention-critic for multi-agent reinforcement learning. In International Conference on Machine Learning, pp. 2961–2970, 2019.
>
> [5] Justin Fu, Aviral Kumar, Ofir Nachum, George Tucker, and Sergey Levine. D4RL: Datasets for deep data-driven reinforcement learning. arXiv preprint arXiv:2004.07219, 2020.

---

> ### Author Response · Authors · 2022-11-13
> **Response to Reviewer E6Nd (Part 1/2)**
>
> Thank you for your inspiring and thorough reviews! We are encouraged by your careful reading and constructive comments. Here are our clarifications for your proposed weaknesses and answers to your questions.
>
> ---
>
> **Weakness 1**: ODIS requires access to global states to learn the skill extractor. It is unclear to me why learning observation encoders and a mixing network, as done in the coordination policy, is insufficient.
>
> **Response 1**: We would like to argue that the observation encoder and mixing network is insufficient. To learn a good coordination policy, an agent needs to infer the other agents' intension. When taking the local observation and the actions of the other agents into consideration, the information of the other agents' policies is incomplete, and the learning is of great difficulty. Involving the other agents' observations drastically helps the learning, since the policy can be reconstructed from the observations and the actions of the other agents. The mixing network cannot help as it does not disclose the other agents' policy but only mix the Q-values of the other agents' actions. Please also note that the joint observation is equivalent to the global state when we consider coordination in the CTDE paradigm.
>
> ---
>
> **Weakness 2**: While Figure 3 seems to support this, learning diverse skills automatically has historically been a difficult problem [1, 2], and I would appreciate some further discussion or ablations around this.
>
> **Response 2**: Our paper is not learning diverse skills. We understand that in previous studies diversity is an important objective for learning skills, which we believe is due to the definition of a skill in the previous single task settings. In this work, we believe the skill can be well reused across different tasks, and therefore, we introduce skill discovery with multi-task data, which is not apparently relevant with the diversity. Thus it is not applicable to study diversity in our situations. We have revised the paper to discuss the relationship with previous diverse skill discovery studies to make this clear.
>
> ---
>
> **Weakness 3**: In most tasks, the results are only slightly better than behavioral cloning.
>
> **Response 3**: We do not think our performance is only slightly better than behavior cloning methods. In Table 1, ODIS outperform all other baselines in about **90%** of tasks and outperforms BC-best by 10% in more than **half of the tasks**. Besides, our behavior cloning baselines, including BC-t and BC-r, adopt the transformer structure and the return-to-go information, which has been found to be effective in offline RL research [3]. Generally, our implemented baselines are strong for fair comparisons.
>
> ---
>
> **Weakness 4**: The transformer-based architecture is closely related to MAAC [4].
>
> **Response 4**: The transformer-based architecture of MAAC is distinguished from our method. MAAC adopts an attention-based centralized critic, but it is not designed for processing alterable agent numbers and observation/action shapes. We think that MAAC is also a related actor-critic MARL algorithm, and we have added it to our related work section.

---

### Author Response · Authors · 2022-11-13
**General Response to All Reviewers**

We appreciate valuable comments from all reviewers. We have revised our paper carefully according to your suggestions. We summarize our modifications as follows.

1. We modify unclear statements in our methodology according to all reviewers' suggestions.
2. We add a citation to MAAC in the related work section (for Reviewer E6Nd).
3. We add more comparisons to single-task offline MARL algorithms in Appendix D (for Reviewer E6Nd).
4. We modify Figure 3 and its discussions to better illustrate our visualization (for Reviewer PuN1).
5. We reorganize the experiments section for better readability (for Reviewer 1rgQ).

The major modifications are colored magenta for the sake of clarity. We hope that our response can address all your concerns of our paper. Please let us know if we miss anything. We are looking forward to further inspiring discussions.

---

### Author Response · Authors · 2022-12-09
**Requesting Feedback on Responses and Revisions**

Dear Reviewers,

We are grateful for the time and effort you took to review our paper and provide detailed feedback. We have submitted our responses and revisions for a few weeks. As the discussion period is coming to a close, we would appreciate it if you could confirm whether there are any remaining issues or questions that need to be addressed. We sincerely welcome further discussions of our paper.

Best regards,

The Authors

---

### Decision · Program_Chairs · 2023-01-20

**Decision:**

Accept: poster

**Justification For Why Not Higher Score:**

Empirical gains could be stronger.

**Justification For Why Not Lower Score:**

Broad consensus that the merits outweigh the weaknesses of this paper.

**Metareview: Summary, Strengths And Weaknesses:**

I thank the authors for their submission and active engagement during the discussion period. The majority of reviewers agree that this paper is worthy of publication. In particular, they found the work to tackle an important problem, the paper well written, the empirical results thorough with appropriate baselines. Reviewer PuN1 had concerns around the methodological novelty. I believe the authors have addressed the concerns of this reviewer well. Therefore, I side with the other reviewers and recommend acceptance.

**Note From Pc:**

if the above contains the word "oral" or "spotlight" please see: "oral" presentation means -> notable-top-5% and "spotlight" means -> notable-top-25%. As stated in our emails, we are disassociating presentation type from AC recommendations